# CONVERGENCE ANALYSIS OF ADAPTIVE GRADIENT METHODS UNDER REFINED SMOOTHNESS AND NOISE ASSUMPTIONS

## ABSTRACT

Adaptive gradient methods, such as AdaGrad, are among the most successful optimization algorithms for neural network training. While these methods are known to achieve better dimensional dependence than stochastic gradient descent (SGD) under favorable geometry for stochastic convex optimization, the theoretical justification for their success in stochastic non-convex optimization remains elusive. In fact, under standard assumptions of Lipschitz gradients and bounded noise variance, it is known that SGD is worst-case optimal (up to absolute constants) in terms of finding a near-stationary point with respect to the $\ell_2$-norm, making further improvements impossible. Motivated by this limitation, we introduce refined assumptions on the smoothness structure of the objective and the gradient noise variance, which better suit the coordinate-wise nature of adaptive gradient methods. Moreover, we adopt the $\ell_1$-norm of the gradient as the stationarity measure, as opposed to the standard $\ell_2$-norm, to align with the coordinate-wise analysis and obtain tighter convergence guarantees for AdaGrad. Under these new assumptions and the $\ell_1$-norm stationarity measure, we establish an *upper bound* on the convergence rate of AdaGrad and a corresponding *lower bound* for SGD. In particular, for certain configurations of problem parameters, we show that the iteration complexity of AdaGrad outperforms SGD by a factor of $d$. To the best of our knowledge, this is the first result to demonstrate a provable gain of adaptive gradient methods over SGD in a non-convex setting. We also present supporting lower bounds, including one specific to AdaGrad and one applicable to general deterministic first-order methods, showing that our upper bound for AdaGrad is tight and unimprovable up to a logarithmic factor under certain conditions.

## 1 INTRODUCTION

Adaptive gradient methods, including variants like AdaGrad (McMahan & Streeter, 2010; Duchi et al., 2011) and Adam (Kingma & Ba, 2015), have become essential for training large-scale neural networks and language models. Their popularity over classic stochastic gradient descent (SGD) (Robbins & Monro, 1951) stems from two key features: (i) adaptive step sizes based on past gradients, eliminating the need for problem-specific parameters like the gradient's Lipschitz constant or stochastic gradient variance, and (ii) the use of coordinate-wise step sizes, allowing better exploitation of the objective's geometry compared to SGD's uniform step size.

Their empirical success has motivated exploring theoretical guarantees that show a provable gain for this class of methods over the traditional SGD method. To pursue this goal, adaptive gradient methods were initially examined in the context of online convex optimization. In particular, it was shown by Duchi et al. (2011) that depending on the geometry of the feasible set and the sparsity of the gradients, AdaGrad's regret bound could be either better or worse than that of SGD by a factor of $\sqrt{d}$, where $d$ represents the problem's dimension. For further details, we refer readers to (Hazan, 2016; Orabona, 2019). Moreover, using the classical online-to-batch conversion (Cesa-Bianchi et al., 2004; Shalev-Shwartz, 2012), these regret bounds directly translate into convergence rate guarantees in stochastic convex optimization.

In the *non-convex setting*, although significant work has been done to characterize the convergence of adaptive methods under various assumptions (more details in the related work section), no provable gain has been established for adaptive methods over SGD, and demonstrating such a gain for AdaGrad in the non-convex setting remains an open problem, see (Chen & Hazan, 2024).

Note that when the objective function is smooth and the stochastic gradients are unbiased with bounded variance, SGD can, after $T$ iterations, find a point where the expected gradient $\ell_2$-norm is bounded by $\mathcal{O}(\frac{1}{T^{1/4}})$ (Ghadimi & Lan, 2013; Bottou et al., 2018). This convergence rate is known to be optimal for any method relying on first-order oracles under the discussed assumptions (Arjevani et al., 2023). Consequently, to demonstrate a provable gain for adaptive methods over SGD in the non-convex setting, we must move beyond the classic setup. In particular, as we will discuss in detail, we argue that modifying both the *assumptions* and the *measure of stationarity* is necessary to better account for the coordinate-wise nature of adaptive methods.

**Contributions.** Motivated by the coordinate-wise structure of AdaGrad, we present refined assumptions on the smoothness and the noise variance by associating each coordinate with a Lipschitz constant $L_i$ and a gradient noise variance $\sigma_i^2$ for $i = 1, 2, \ldots, d$ (see Assumptions 2.3b and 2.4b). However, even under these refined assumptions, we show that SGD is still worst-case optimal in the noiseless setting when the $\ell_2$-norm is the measure of stationarity (Theorem 2.1). Thus, we change the measure of stationarity to the $\ell_1$-norm and demonstrate that, with these new assumptions and the revised stationarity measure, it is possible to prove that AdaGrad achieves an upper bound complexity that outperforms the lower bound complexity for SGD. Our main contributions are summarized below:

- **Upper bound for AdaGrad:** Let $\boldsymbol{L} = [L_1, \ldots, L_d] \in \mathbb{R}^d$ and $\boldsymbol{\sigma} = [\sigma_1, \ldots, \sigma_d] \in \mathbb{R}^d$ denote the Lipschitz constant vector and the noise variance vector, respectively. We establish that AdaGrad achieves a rate of $\mathcal{O}\left(\sqrt{\frac{\|\boldsymbol{L}\|_1 \log h(T)}{T}} + \left(\frac{\|\boldsymbol{\sigma}\|_1^2 \|\boldsymbol{L}\|_1 \log h(T)}{T}\right)^{1/4} + \frac{\|\boldsymbol{\sigma}\|_1 \sqrt{\log h(T)}}{T^{1/4}}\right)$ in terms of the $\ell_1$-norm, where $h(T)$ is a polynomial function of $T$ and $d$ (Theorem 3.1). Notably, this rate depends on $d$ only implicitly through $\boldsymbol{L}$ and $\boldsymbol{\sigma}$.

- **Lower bound for SGD:** Under the same assumptions and using the $\ell_1$-norm as the stationarity measure, we show that the convergence rate of SGD with a constant step size is lower bounded by $\Omega\left(\sqrt{\frac{d\|\boldsymbol{L}\|_\infty}{T}} + \frac{d^{1/4}\left(\sum_{i=1}^d \sigma_i \sqrt{L_i}\right)^{1/2}}{T^{1/4}}\right)$ when the number of iterations $T$ is sufficiently large (Theorem 4.1).

- **Provable gain for AdaGrad over SGD:** By comparing AdaGrad's upper bound with SGD's lower bound, we show that when the parameters $\boldsymbol{L}$ and $\boldsymbol{\sigma}$ are both sparse and aligned in a certain way, AdaGrad's complexity can be $d$ times better than the one for SGD.

- **Lower bounds for AdaGrad**: We establish a complexity lower bound for AdaGrad, matching the first term in our upper bound up to absolute constants (including the $\log T$ factor), as well as the second term under certain conditions on $\boldsymbol{L}$ and $\boldsymbol{\sigma}$ (Theorem 2.1). We also provide a lower bound of $\Omega\left(\sqrt{\frac{\|\boldsymbol{L}\|_1}{T}}\right)$ for all deterministic first-order methods in the noiseless case, showing the first term is unimprovable up to log factors (Theorem 3.3).

## 1.1 Related work

**AdaGrad-Norm.** Several prior works have established that AdaGrad-Norm achieves a convergence rate similar to that of SGD, but under stronger assumptions, such as bounded gradients (Ward et al., 2020; Kavis et al., 2022; Gadat & Gavra, 2022), the step-size being (conditionally) independent of the stochastic gradient (Li & Orabona, 2019; 2020), or sub-Gaussian noise (Li & Orabona, 2020; Kavis et al., 2022). Faw et al. (2022) addressed this issue and showed that under standard assumptions—Lipschitz gradients and bounded variance—AdaGrad-Norm achieves the same complexity as SGD in terms of gradient's $\ell_2$-norm (up to a logarithmic factor). They further explored the setting where the stochastic gradient has affine variance. In addition, several works (Attia & Koren, 2023; Liu et al., 2023) provided high-probability convergence guarantees for AdaGrad-Norm under sub-Gaussian noise assumptions. The extension to the generalized smoothness setting (Zhang et al., 2020) was developed in Faw et al. (2023); Wang et al. (2023). However, as mentioned earlier, these results do not demonstrate any improvement over SGD in terms of convergence rate.

**AdaGrad and its variants.** Most works on AdaGrad and its variants, such as RMSProp (Tieleman & Hinton, 2012), Adam (Kingma & Ba, 2015) and AMSGrad (Reddi et al., 2018), employed the gradient $\ell_2$-norm as the stationarity measure. Under the assumption of bounded gradients, Chen et al. (2019); Alacaoglu et al. (2020); Défossez et al. (2022) established a rate of $\mathcal{O}(\frac{1}{T^{1/4}})$, but with an explicit dimension dependence of at least $\Omega(d^{1/4})$. Thus, these convergence results could be worse than the dimensional-free rate of SGD. Recently, several papers have studied the convergence of adaptive methods with respect to the gradient's $\ell_1$-norm, closely related to our work. Under the assumption of coordinate-wise subgaussian noise, Liu et al. (2023) provided a high-probability rate for AdaGrad of $\tilde{\mathcal{O}}\left(\frac{d}{\sqrt{T}} + \frac{d}{T^{1/4}}\right)$, which is worse than our worst-case rate by a factor of $\sqrt{d}$. Li & Lin (2024) analyzed RMSProp under the standard smoothness assumption and a coordinate-wise bounded noise variance assumption and showed a convergence rate of $\tilde{\mathcal{O}}(\frac{\sqrt{d}}{\sqrt{T}} + \frac{\sqrt{d}}{T^{1/4}})$, which matches our worst-case bound. However, their convergence result only showed the possibility of matching the convergence rate of SGD instead of surpassing it, and thus it did not fully explain the advantage of adaptive gradient methods. Along a different line of research, Crawshaw et al. (2022) proposed a generalized SignSGD algorithm and analyzed its rate in terms of the gradient's $\ell_1$-norm, under their proposed coordinate-wise generalized smoothness and subgaussian noise assumptions. However, their results are not directly comparable to ours due to the different assumptions and algorithms.

**Lower bounds.** Several works have studied the complexity of finding an $\epsilon$-stationary point of a smooth non-convex optimization with exact or noisy gradient oracles. However, to the best of our knowledge, they all use the $\ell_2$-norm of the gradient as the stationarity measure. In the noiseless setting, Carmon et al. (2020) showed that all first-order methods require at least $\Omega(\frac{1}{\epsilon^2})$ gradient queries for finding a point $x$ with $\|\nabla f(x)\|_2 \le \epsilon$. Building on similar techniques, Arjevani et al. (2023) extended it to non-convex stochastic optimization and showed a lower bound of $\Omega(\frac{1}{\epsilon^4})$ for finding a point $x$ with $\mathbb{E}[\|\nabla f(x)\|_2] \le \epsilon$. In addition to the use of $\ell_2$-norm, these works focus on establishing dimensional-free lower bounds and the constructed worst-case instance has a dimension that grows with $1/\epsilon$. As a result, their techniques are unfit for studying lower bounds in a given dimension, which is our focus here. Along a different line of work, people have studied the complexity of finding $\epsilon$-stationary points of a function in a small dimension (Vavasis, 1993; Cartis et al., 2010; Chewi et al., 2023). In particular, Chewi et al. (2023) showed that any deterministic first-order method would require $\Omega(\frac{1}{\epsilon^2})$ to find the $\epsilon$-stationary point of a one-dimensional smooth non-convex function. To the best of our knowledge, our result is the first to establish a lower bound in terms of the $\ell_1$-norm and highlight the dimensional dependence in the convergence rate.

**Concurrent work.** The concurrent work by Liu et al. (2024), which appeared online two weeks after our initial paper, also examined AdaGrad's convergence under anisotropic smoothness and noise assumptions, similar to our refined Assumptions 2.3b and 2.4b. They proved an upper bound on AdaGrad's convergence rate in terms of the gradient's $\ell_1$-norm, comparable to our result in Theorem 3.1, and compared it with the classical upper bound for SGD in terms of the $\ell_2$-norm. In contrast, our approach focuses on establishing a lower bound for SGD, allowing us to directly compare AdaGrad's upper bound with SGD's lower bound to demonstrate a clear advantage for AdaGrad. Moreover, we further validate the tightness of our AdaGrad upper bound through two lower bounds, one specific to AdaGrad and another for deterministic first-order methods.

## 2 PRELIMINARIES

**Notation.** We use boldface letters for vectors and normal font letters for scalars. The Euclidean or $\ell_2$-norm of a vector $\boldsymbol{w}$ is denoted by $\|\boldsymbol{w}\|_2$ and its $\ell_1$ norm is indicated by $\|\boldsymbol{w}\|_1$. For a vector $\boldsymbol{w} \in \mathbb{R}^d$, we denote its $i$-th coordinate by $w_i$. We use $[n]$ to denote the set $\{1, 2, \ldots, n\}$. Further, $\mathcal{F}_t$ denotes the $\sigma$-algebra generated after time index $t$. In our case, $\mathcal{F}_t$ contains all iterates $\boldsymbol{w}_0, \ldots, \boldsymbol{w}_{t+1}$ and all stochastic gradients $\boldsymbol{g}_0, \ldots, \boldsymbol{g}_t$. Finally, the notation $\tilde{\mathcal{O}}$ suppresses logarithmic dependencies.

In this paper, our objective is to identify an approximate stationary point of a smooth, non-convex function $F : \mathbb{R}^d \to \mathbb{R}$ over the unbounded domain $\mathbb{R}^d$. The most commonly analyzed AdaGrad-type method in the literature is AdaGrad-Norm, which was first considered in McMahan & Streeter (2010). Specifically, AdaGrad-Norm updates the iterates $\boldsymbol{w}_t$ according to the following update rule:

$$\boldsymbol{w}_{t+1} = \boldsymbol{w}_t - \frac{\eta}{b_t + \delta}\,\boldsymbol{g}_t, \quad \text{where} \quad b_t = \sqrt{\sum_{s=1}^{t} \|\boldsymbol{g}_s\|^2}, \qquad \text{(AdaGrad-Norm)}$$

where $\boldsymbol{g}_t$ is the stochastic gradient of $F$ at $\boldsymbol{w}_t$, the scalar $\eta$ is a scaling parameter, $\delta > 0$ is a small constant to ensure numerical stability. However, as mentioned in the introduction, most prior works demonstrated convergence similar to the guarantees obtained by SGD. In this paper, we focus on the coordinate-wise variant of AdaGrad, whose updates are given by

$$w_{t+1,i} = w_{t,i} - \eta \frac{g_{t,i}}{b_{t,i} + \delta}, \quad \text{where} \quad b_{t,i} = \sqrt{\sum_{s=1}^{t} g_{s,i}^2} \ \forall i \in [d], \qquad \text{(AdaGrad)}$$

where constant $\delta$ is introduced to ensure numerical stability. Some literature refers to this algorithm as "diagonal AdaGrad" or "coordinate-wise AdaGrad", while reserving the name AdaGrad for the variant involving full matrix inversion. In this work, we refer to the diagonal version as AdaGrad, as it is the most widely used in practice.

## 2.1 Assumptions and measure of stationarity

In this section, we outline the assumptions required to characterize the complexity of AdaGrad. To provide motivation, we first revisit the standard assumptions on the objective function $F$ and its stochastic gradient, which are commonly used in the analysis of stochastic first-order methods (Ghadimi & Lan, 2013; Bottou et al., 2018).

**Assumption 2.1.** *The function $F(\cdot)$ is bounded from below, i.e., $\inf_{\boldsymbol{w} \in \mathbb{R}^d} F(\boldsymbol{w}) = F^* > -\infty$.*

**Assumption 2.2.** *The stochastic gradient $\boldsymbol{g}_t$ is unbiased, i.e., $\mathbb{E}[\boldsymbol{g}_t \mid \mathcal{F}_{t-1}] = \nabla F(\boldsymbol{w}_t)$.*

**Assumption 2.3a.** *The stochastic gradient $\boldsymbol{g}_t$ has a bounded variance, i.e., $\mathbb{E}[\|\boldsymbol{g}_t - \nabla F(\boldsymbol{w}_t)\|^2] \leq \sigma^2$ for some non-negative constant $\sigma$.*

**Assumption 2.4a.** *The function $F(\cdot)$ is smooth, i.e., for any vectors $\boldsymbol{x}, \boldsymbol{y} \in \mathbb{R}^d$, we have $|F(\boldsymbol{x}) - F(\boldsymbol{y}) - \langle \nabla F(\boldsymbol{x}), \boldsymbol{x} - \boldsymbol{y} \rangle| \leq \frac{L}{2} \|\boldsymbol{x} - \boldsymbol{y}\|^2$, where $L \geq 0$ is the Lipschitz constant of the gradient of $F$.*

Under Assumptions 2.1-2.4a, it is known that SGD, with an appropriately chosen step size, can find a point $\hat{\boldsymbol{w}}$ such that $\mathbb{E}\left[\|\nabla F(\hat{\boldsymbol{w}})\|_2^2\right] \leq \epsilon^2$ after at most $\mathcal{O}\left(\frac{L(F(\boldsymbol{w}_1) - F^*)\sigma^2}{\epsilon^4} + \frac{(F(\boldsymbol{w}_1) - F^*)L}{\epsilon^2}\right)$ iterations (Ghadimi & Lan, 2013; Bottou et al., 2018). Moreover, this complexity matches the lower bound for any first-order method up to an absolute constant, as shown by Arjevani et al. (2023).

According to this classical convergence theory, SGD is the optimal first-order method in this setting in the worst-case sense, leaving no room for further improvement. However, coordinate-wise adaptive methods, such as AdaGrad, are often observed to converge significantly faster than SGD in practice. Intuitively, the main advantage of AdaGrad over SGD is that each coordinate employs a different step size that adapts to the gradients of each respective coordinate. In contrast, SGD uses the same step size across all coordinates, and thus its step size is constrained by the most "difficult" coordinate, impeding progress in other coordinates that could allow a larger step size. Consequently, we expect AdaGrad to outperform SGD when the coordinates exhibit imbalance. To better capture how coordinate-wise AdaGrad exploits structural features, we propose replacing Assumptions 2.3a and 2.4a with their coordinate-wise refined counterparts, inspired by Bernstein et al. (2018).

**Assumption 2.3b.** *The stochastic gradient $\boldsymbol{g}_t$ with elements $[g_{t,1}, \ldots, g_{t,d}]$ has a coordinate-wise bounded variance. That is, for all $i \in [d]$, we have $\mathbb{E}[|g_{t,i} - \nabla_i F(\boldsymbol{w}_t)|^2 \mid \mathcal{F}_{t-1}] \leq \sigma_i^2$, where $\sigma_i$ is a non-negative constant and $\nabla_i F(\boldsymbol{w}_t)$ represents the $i$-th coordinate of the gradient $\nabla F(\boldsymbol{w}_t)$. Moreover, we define the vector $\boldsymbol{\sigma}$ as $\boldsymbol{\sigma} = [\sigma_1, \sigma_2, .., \sigma_d] \in \mathbb{R}^d$.*

The above condition on the variance of the stochastic gradient is a more fine-grained assumption compared to the standard assumption. Indeed, our considered assumption implies Assumption 2.3a when we consider $\sigma^2 = \sum_{i=1}^d \sigma_i^2$. As discussed earlier, since we aim to study an algorithm with a coordinate-specific update, the above assumption better captures its convergence behavior.

**Assumption 2.4b.** *The function $F(\cdot)$ is coordinate-wise smooth, i.e., $\forall \boldsymbol{x}, \boldsymbol{y} \in \mathbb{R}^d$, $|F(\boldsymbol{y}) - F(\boldsymbol{x}) - \langle \nabla F(\boldsymbol{x}), \boldsymbol{y} - \boldsymbol{x} \rangle| \leq \sum_{i=1}^d \frac{L_i}{2} |x_i - y_i|^2$, where the constant $L_i > 0$ is the Lipschitz constant associated with the $i$-th coordinate. Moreover, we define the vector $\boldsymbol{L}$ as $\boldsymbol{L} = [L_1, L_2, .., L_d] \in \mathbb{R}^d$.*

Assumption 2.4b is similar to the fine-grained assumptions made in the literature for coordinate-wise analysis of algorithms Richtárik & Takáč (2011); Bernstein et al. (2018). We recover the standard smoothness in Assumption 2.4a by considering the Lipschitz constant as $L := \max_i L_i = \|\boldsymbol{L}\|_\infty$.

Besides the assumptions, the choice of stationarity measure is crucial in characterizing an algorithm's complexity. In non-convex optimization, the standard choice is the Euclidean $\ell_2$-norm of the gradient. However, this choice may be inadequate to demonstrate the advantage of AdaGrad over SGD. To illustrate this, consider the noiseless setting where $\sigma_i = 0$ for all $i \in [d]$ and thus SGD reduces to gradient descent. Under Assumption 2.4b, the gradient of $F$ is $\|\boldsymbol{L}\|_\infty$-Lipschitz, and standard analysis shows that gradient descent with step size $\eta = 1/\|\boldsymbol{L}\|_\infty$ can find a point $\hat{\boldsymbol{w}}$ such that $\|\nabla F(\hat{\boldsymbol{w}})\|_2 \leq \epsilon$ after at most $\frac{2\|\boldsymbol{L}\|_\infty (F(\boldsymbol{w}_1) - F^*)}{\epsilon^2}$ iterations. The following theorem shows that if the $\ell_2$-norm of the gradient is used as the stationarity measure, no deterministic first-order method can outperform gradient descent by more than a factor of two, even under the refined Assumption 2.4b.

**Theorem 2.1.** *Consider any deterministic algorithm $\mathcal{A}$ with only access to the first-order oracle with an initial point $\boldsymbol{x}_1 \in \mathbb{R}^d$. For any positive vector $\mathbf{L} = [L_1, \ldots, L_d]$ and any $\Delta_f > 0$, there exists a function $f : \mathbb{R}^d \to \mathbb{R}$ such that: (i) $f$ satisfies Assumption 2.4b and $f(\boldsymbol{x}_1) - \inf f \leq \Delta_f$; (ii) Algorithm $\mathcal{A}$ requires more than $\frac{\|\mathbf{L}\|_\infty \Delta_f}{\epsilon^2}$ gradient queries to find a point $\hat{\mathbf{x}}$ with $\|\nabla f(\hat{\mathbf{x}})\|_2 < \epsilon$.*

*Proof sketch.* Inspired by similar arguments in Chewi et al. (2023), we employ the concept of a "resisting oracle" (Nemirovski & Yudin, 1983; Nesterov, 2018) in our proof. Specifically, consider any deterministic method $\mathcal{A}$ that has access only to a first-order oracle, and let $T$ be an integer satisfying $T \leq \frac{\|\mathbf{L}\|_\infty \Delta_f}{\epsilon^2}$. We will adversarially construct a function $f$ that satisfies the stated requirements and ensures that $\nabla f(\boldsymbol{x}_t) = [\epsilon, 0, 0, \ldots, 0] \in \mathbb{R}^d$ for any $t \in [T]$, where $\{\boldsymbol{x}_t\}_{t=1}^T$ are the queries made by $\mathcal{A}$. Crucially, the function $f$ is not fixed in advance but is built based on the points $\boldsymbol{x}_1, \boldsymbol{x}_2, \ldots, \boldsymbol{x}_T$ queried by $\mathcal{A}$. This is possible due to the deterministic nature of $\mathcal{A}$, which allows us to "simulate" the algorithm using the known responses from the first-order oracle. Hence, we only need to show that there exists a function $f$ that satisfies the stated properties and is consistent with the output provided by the resisting oracle.

Without loss of generality, assume $L_1 = \|\boldsymbol{L}\|_\infty$. We construct the adversarial function in the form of $f(\boldsymbol{x}) = \Delta_f p(\sqrt{L_1/\Delta_f} x^{(1)})$, where $x^{(1)}$ is the first coordinate of $\boldsymbol{x}$ and $p : \mathbb{R} \to \mathbb{R}$ is a function of one dimension to be determined. Let $\{x_t^{(1)}\}_{t=1}^T$ be the first coordinate of the queries $\{\boldsymbol{x}_t\}_{t=1}^T$. Since $T \leq \frac{\|\boldsymbol{L}\|_\infty \Delta_f}{\epsilon^2}$, by invoking Lemma C.1 in Appendix C.1, we show the existence of a function $p$ satisfying the following conditions: (i) its gradient $p'$ is 1-Lipschitz; (ii) $p(\sqrt{\frac{L_1}{\Delta_f}} x_1^{(1)}) - \inf p \leq 1$; (iii) $p'(\sqrt{\frac{L_1}{\Delta_f}} x_t^{(1)}) = \frac{\epsilon}{\sqrt{L_1 \Delta_f}}$ for any $t \in [T]$. It is easy to verify that $f$ meets all the required assumptions, and $\forall t \in [T], \|\nabla f(\boldsymbol{x}_t)\|_2 = |\sqrt{L_1 \Delta_f} p'(\sqrt{\frac{L_1}{\Delta_f}} x_t^{(1)})| = \epsilon$. The proof is complete. □

The lower bound in Theorem 2.1 matches the upper bound of SGD (up to a constant factor of 2), which certifies the optimality of SGD with respect to the gradient $\ell_2$-norm. To provide some intuition for this result, note that in the proof of Theorem 2.1, the worst-case function for any deterministic first-order method can be realized by a function $f$ that is effectively one-dimensional. As such, the complexity bound does not reflect the imbalance between different coordinates. This observation motivates the use of an alternative stationarity measure. As we will demonstrate in the next section, the convergence analysis suggests that the gradient $\ell_1$-norm is a more suitable choice for AdaGrad.

## 3 $\ell_1$-NORM CONVERGENCE OF ADAGRAD: UPPER AND LOWER BOUNDS

In this section, we present our main convergence results for AdaGrad. In Section 3.1, we derive an upper bound on the number of iterations required to find a near-stationary point in terms of the $\ell_1$-norm, instead of the conventional $\ell_2$-norm. As discussed earlier, this stationarity measure is more suitable given the coordinate-specific structure of AdaGrad and better highlights the advantages compared to SGD and AdaGrad-Norm, as we will demonstrate. Then in Section 3.2, we provide supporting lower bounds to demonstrate that our upper bounds are tight under specific settings.

### 3.1 UPPER BOUND

In this section, we first state our main convergence result for AdaGrad in terms of the expected average $\ell_1$-norm of the gradient. Due to space limitations, we provide a proof sketch below and the complete proof can be found in Appendix B.

**Theorem 3.1.** *Let $\{\boldsymbol{w}_t\}_{t=1}^{T}$ be the iterates generated by AdaGrad with $\delta < \frac{1}{d}$ and suppose that Assumptions 2.1, 2.2, 2.3b, and 2.4b hold. Then $\mathbb{E}\left[\frac{1}{T}\sum_{t=1}^{T}\|\nabla F(\boldsymbol{w}_t)\|_1\right]$ is upper bounded by*

$$\mathcal{O}\left(\frac{\Delta_F}{\eta\sqrt{T}}+\frac{\eta\|\boldsymbol{L}\|_1\log h(T)}{\sqrt{T}}+\frac{\sqrt{\|\boldsymbol{\sigma}\|_1\Delta_F}}{\sqrt{\eta}T^{\frac{1}{4}}}+\frac{\sqrt{\eta\|\boldsymbol{\sigma}\|_1\|\boldsymbol{L}\|_1\log h(T)}}{T^{\frac{1}{4}}}+\frac{\|\boldsymbol{\sigma}\|_1\sqrt{\log h(T)}}{T^{\frac{1}{4}}}\right),\quad(1)$$

*where $\Delta_F = F(\boldsymbol{w}_1) - F^*$ and $h(T) = \mathcal{O}\left(\frac{T\|\boldsymbol{\sigma}\|_\infty^2+T\|\nabla F(\boldsymbol{w}_1)\|_\infty^2+\eta^2\|\boldsymbol{L}\|_\infty\|\boldsymbol{L}\|_1 T^3}{\delta^2}\right)$.*

*Proof Sketch.* Our proof consists of the following steps.

**Step 1:** Define $\eta_{t,i} = \frac{\eta}{b_{t,i}+\delta}$ and rewrite AdaGrad as $w_{t+1,i} = w_{t,i} - \eta_{t,i}g_{t,i}$. By applying Assumption 2.4b to two consecutive iterates $\boldsymbol{w}_t$ and $\boldsymbol{w}_{t+1}$, we obtain the descent inequality $F(\boldsymbol{w}_{t+1}) \le F(\boldsymbol{w}_t) - \sum_{i=1}^{d}\eta_{t,i}g_{t,i}\nabla_i F(\boldsymbol{w}_t) + \sum_{i=1}^{d}\frac{L_i}{2}\eta_{t,i}^2 g_{t,i}^2$. Note that $\eta_{t,i}$ and $g_{t,i}$ are correlated and thus $\mathbb{E}[\eta_{t,i}g_{t,i}\mid\mathcal{F}_{t-1}] \neq \eta_{t,i}\mathbb{E}[g_{t,i}\mid\mathcal{F}_{t-1}]$, which is one of the main challenges of analyzing adaptive gradient methods. To address this, following (Ward et al., 2020; Faw et al., 2022), we introduce a "decorrelated step size" as:

$$\hat{\eta}_{t,i} = \frac{\eta}{\sqrt{b_{t-1,i}^2+\sigma_i^2+\nabla_i F(\boldsymbol{w}_t)^2}+\delta}.\quad(2)$$

Compared to the definition $\eta_{t,i} = \frac{\eta}{\sqrt{b_{t-1,i}^2+g_{t,i}^2}+\delta}$, the stochastic gradient $g_{t,i}^2$ is replaced with $\nabla_i F(\boldsymbol{w}_t)^2 + \sigma_i^2$ in (2) and as a result $\hat{\eta}_{t,i}$ and $g_{t,i}$ are independent conditioned on $\mathcal{F}_{t-1}$. Using the decorrelated step size, we obtain the following key inequality (see Corollary B.3):

$$\mathbb{E}\left[\sum_{t=1}^{T}\sum_{i=1}^{d}\frac{\hat{\eta}_{t,i}}{2}\nabla_i F(\boldsymbol{w}_t)^2\right] \le F(\boldsymbol{w}_1) - F^* + \left(2\eta\|\boldsymbol{\sigma}\|_1+\frac{\eta^2\|\boldsymbol{L}\|_1}{2}\right)\log h(T),\quad(3)$$

where $h(T) = 1 + \frac{T\|\boldsymbol{\sigma}\|_\infty^2}{\delta^2} + \frac{T(\|\nabla F(\boldsymbol{w}_1)\|_\infty+\eta\sqrt{\|\boldsymbol{L}\|_\infty\|\boldsymbol{L}\|_1 T})^2}{\delta^2}$.

**Step 2:** In light of (3), it remains to establish lower bounds on the step sizes $\hat{\eta}_{t,i}$. Since each coordinate is updated independently, we study each coordinate and construct a uniform lower bound on $\hat{\eta}_{t,i}$ for $t \in [T]$. Specifically, for each $i \in [d]$, we define a new auxiliary step size $\tilde{\eta}_{T,i}$ as

$$\tilde{\eta}_{T,i} = \frac{\eta}{\sqrt{\sum_{i=1}^{T-1}g_{t,i}^2+\sum_{t=1}^{T}\nabla_i F(\boldsymbol{w}_t)^2+\sigma_i^2}+\delta}.\quad(4)$$

From (2) and $b_{t-1,i} = \sum_{s=1}^{t-1}g_{s,i}^2$ in AdaGrad, it can be shown that $\hat{\eta}_{t,i} \ge \tilde{\eta}_{T,i}$ for all $t \in [T]$. Moreover, we separate the step sizes from the gradients as follows:

$$\mathbb{E}\left[\sum_{t=1}^{T}\frac{\hat{\eta}_{t,i}}{2}\nabla_i F(\boldsymbol{w}_t)^2\right] \ge \mathbb{E}\left[\frac{\tilde{\eta}_{T,i}}{2}\sum_{t=1}^{T}\nabla_i F(\boldsymbol{w}_t)^2\right] \ge \mathbb{E}\left[\sqrt{\sum_{t=1}^{T}\nabla_i F(\boldsymbol{w}_t)^2}\right]^2 \times \frac{1}{\mathbb{E}\left[\frac{2}{\tilde{\eta}_{T,i}}\right]},\quad(5)$$

where we used that $\mathbb{E}\left[\frac{X^2}{Y}\right] \ge \frac{\mathbb{E}[X]^2}{\mathbb{E}[Y]}$ for any two positive random variables $X$ and $Y$. Hence, we proceed to establish an upper bound on $\mathbb{E}\left[\frac{1}{\tilde{\eta}_{T,i}}\right]$ (see Lemma B.4):

$$\mathbb{E}\left[\frac{1}{\tilde{\eta}_{T,i}}\right] \le \frac{\sigma_i\sqrt{2T}+\delta}{\eta} + \frac{\sqrt{3}\mathbb{E}\left[\sqrt{\sum_{t=1}^{T}\nabla_i F(\boldsymbol{w}_t)^2}\right]}{\eta}.\quad(6)$$

**Step 3:** Note that the upper bound in (6) depends on the sum $\mathbb{E}\left[\sqrt{\sum_{t=1}^{T}\nabla_i F(\boldsymbol{w}_t)^2}\right]$, which also appears on the right hand side of (5). By combining (3), (5) and (6), we arrive at (see Lemma B.5):

$$\mathbb{E}\left[\sum_{i=1}^{d}\sqrt{\sum_{t=1}^{T}\nabla_i F(\boldsymbol{w}_t)^2}\right] \le \frac{2\sqrt{3}}{\eta}Q + \sqrt{\frac{2d\delta Q}{\eta}} + 2\sqrt{\frac{\|\boldsymbol{\sigma}\|_1 Q}{\eta}}T^{\frac{1}{4}},\quad(7)$$

where $Q$ denotes the right-hand side of (3). The last step is to relate the left-hand side of the inequality in (7) to the $\ell_1$-norm of the gradients. Specifically, we can write:

$$\frac{1}{T}\sum_{t=1}^{T}\|\nabla F(\boldsymbol{w}_t)\|_1 = \frac{1}{T}\sum_{t=1}^{T}\sum_{i=1}^{d}|\nabla_i F(\boldsymbol{w}_t)| = \frac{1}{T}\sum_{i=1}^{d}\sum_{t=1}^{T}|\nabla_i F(\boldsymbol{w}_t)| \leq \frac{1}{\sqrt{T}}\sum_{i=1}^{d}\sqrt{\sum_{t=1}^{T}|\nabla_i F(\boldsymbol{w}_t)|^2},$$

where we switched the order of the two summations in the second equality and used the Cauchy-Schwarz inequality in the last inequality. This leads to our main theorem. $\qquad\square$

**Remark 3.1.** *We observe that the $\ell_1$-norm of the gradient naturally emerges as the convergence measure, as it provides* the tightest bound *derivable from the inequality in Lemma B.5. Indeed, the $\ell_1$-norm is always an upper bound on the $\ell_2$-norm, and thus the above bound also immediately implies an upper bound on $\frac{1}{T}\sum_{t=1}^{T}\|\nabla F(\boldsymbol{w}_t)\|_2$. However, this relaxation will undermine the advantage of AdaGrad when compared to SGD or AdaGrad-Norm.*

A few remarks on Theorem 3.1 are in order. First, a key feature of the upper bound in (1) is that, apart from the logarithmic term $\log h(T)$, it does not explicitly depend on the dimension $d$. Instead, the dependence is implicit via the variance vector $\boldsymbol{\sigma}$ and the Lipschitz vector $\boldsymbol{L}$ defined in Assumptions 2.3b and 2.4b. In contrast, as shown later in Section 4, SGD unavoidably will incur an explicit dependence on the dimension $d$ in its convergence bound. Moreover, if we select the scaling parameter $\eta$ in AdaGrad to achieve the best convergence bound, then (1) will become

$$\mathcal{O}\left(\sqrt{\frac{\|\boldsymbol{L}\|_1\Delta_F\log h(T)}{T}} + \left(\frac{\|\boldsymbol{\sigma}\|_1^2\|\boldsymbol{L}\|_1\Delta_F\log h(T)}{T}\right)^{1/4} + \frac{\|\boldsymbol{\sigma}\|_1\sqrt{\log h(T)}}{T^{1/4}}\right). \qquad (8)$$

This bound is adaptive to the noise level: when the noise level in the stochastic gradient is relatively small, i.e., $\|\boldsymbol{\sigma}\|_1^2 \ll \frac{\|\boldsymbol{L}\|_1\Delta_F}{T}$, then AdaGrad will achieve a faster rate of $\mathcal{O}(\sqrt{\frac{\|\boldsymbol{L}\|_1\Delta_F\log h(T)}{T}})$. As shown in the next section, this rate matches our lower bound in the noiseless case, up to a log factor.

To aid our discussions and comparisons with existing results, we rewrite our bound in terms of the gradient's Lipschitz constants and the gradient noise variance as in Assumptions 2.3a and 2.4a, commonly used in the literature. Specifically, Assumption 2.3b implies that $\mathbb{E}\left[\|\boldsymbol{g}_t - \nabla F(\boldsymbol{w}_t)\|_2^2\right] \leq \sum_{i=1}^{d}\sigma_i^2 = \|\boldsymbol{\sigma}\|_2^2$ and Assumption 2.4b implies that the function $F$ is $\|\boldsymbol{L}\|_\infty$-Lipschitz. Thus, when we translate our bounds to the standard assumptions that are not tailored for coordinate-wise analysis, the ratios of $\frac{\|\boldsymbol{L}\|_1}{\|\boldsymbol{L}\|_\infty}$ and $\frac{\|\boldsymbol{\sigma}\|_1}{\|\boldsymbol{\sigma}\|_2}$ appear in the upper bound. Given the behavior of these ratios, the dependence of our final bound on $d$ could change, as described in the following cases:

- **Worst case:** In this case, we have $\frac{\|\boldsymbol{L}\|_1}{\|\boldsymbol{L}\|_\infty} = \Theta(d)$ and $\frac{\|\boldsymbol{\sigma}\|_1}{\|\boldsymbol{\sigma}\|_2} = \Theta(\sqrt{d})$. Then the bound in (8) reduces to $\tilde{\mathcal{O}}\left(\sqrt{\frac{d\|\boldsymbol{L}\|_\infty\Delta_F}{T}} + \sqrt{d}\left(\frac{\|\boldsymbol{\sigma}\|_2^2\|\boldsymbol{L}\|_\infty\Delta_F}{T}\right)^{1/4} + \frac{\sqrt{d}\|\boldsymbol{\sigma}\|_2}{T^{1/4}}\right)$. Focusing on the dependence on the dimension $d$, we obtain the rate of $\tilde{\mathcal{O}}(\frac{\sqrt{d}}{\sqrt{T}} + \frac{\sqrt{d}}{T^{1/4}})$.

- **Well-structured case:** In this case, we have $\frac{\|\boldsymbol{L}\|_1}{\|\boldsymbol{L}\|_\infty} = \mathcal{O}(1)$ and $\frac{\|\boldsymbol{\sigma}\|_1}{\|\boldsymbol{\sigma}\|_2} = \mathcal{O}(1)$. This indicates that the curvature and gradient noise are heterogeneous and primarily influenced by a few dominant coordinates. Under such circumstances, our convergence rate in (8) becomes a dimensional-independent rate of $\tilde{\mathcal{O}}(\frac{1}{\sqrt{T}} + \frac{1}{T^{1/4}})$.

We also present a detailed comparison with the existing results for AdaGrad in Appendix A.

## 3.2 LOWER BOUNDS

After establishing an upper bound for AdaGrad, we move on to show a lower bound under the same conditions. For simplicity, we set $\delta = 0$ in AdaGrad, but generalizing to $\delta > 0$ is straightforward.

**Theorem 3.2.** *Consider running AdaGrad with $\delta = 0$ and the scaling parameter $\eta$. Let $\mathbf{L} = [L_1, L_2, \ldots, L_d]$, $\boldsymbol{\sigma} = [\sigma_1, \sigma_2, \ldots, \sigma_d]$ and $\Delta_f > 0$ be given parameters. Then there exists a function $f : \mathbb{R}^d \to \mathbb{R}$ such that: (i) $f$ satisfies Assumption 2.4b and $f(\boldsymbol{x}_1) - \inf f \leq \Delta_f$; (ii) The stochastic gradient $\boldsymbol{g}_t$ satisfies Assumptions 2.2 and 2.3b; (iii) We have $\mathbb{E}\left[\min_{1 \leq t \leq T}\|\nabla f(\mathbf{x}_t)\|_1\right] = \Omega\left(\max\left\{\sqrt{\frac{\|\mathbf{L}\|_1\Delta_f\log T}{T}}, \left(\frac{(\sum_{i=1}^{d}\sigma_i^{2/3}L_i^{1/3})^3\Delta_f\log T}{T}\right)^{\frac{1}{4}}\right\}\right).$*

*Proof Sketch.* We construct the function $f$ in the form of $f(\boldsymbol{x}) = \sum_{i=1}^{d} p_i(x^{(i)})$, where $x^{(i)}$ denotes the $i$-th coordinate of the vector $\boldsymbol{x} \in \mathbb{R}^d$ and $p_i : \mathbb{R} \to \mathbb{R}$ is a one-dimensional function to be specified. Since each coordinate is updated independently in AdaGrad, this is equivalent to running AdaGrad on each of the one-dimensional functions $p_i$ in parallel. Thus, this requires us to understand the convergence lower bound for AdaGrad in the one-dimensional setting.

In one dimension, AdaGrad follows the update rule $x_{t+1} = x_t - \frac{\eta}{\sqrt{\sum_{s=1}^{t} |g_s|^2}} g_t$, where $g_t$ denotes the stochastic gradient at time step $t$. In Corollary C.4, we will show that there exists a one-dimensional function $p_{\Delta,L,\sigma,T}(\cdot)$ and a stochastic gradient oracle such that: (i) Its gradient is $L$-Lipschitz and its initial function value gap is bounded by $\Delta$; (ii) The stochastic gradient oracle in unbiased with bounded variance $\sigma^2$; (iii) The iterates of AdaGrad after $T$ iterations satisfy $\mathbb{E}\left[\min_{1 \le t \le T} |p'(x_t)|\right] = \Omega(\sqrt{\frac{L\Delta \log T}{T}} + (\frac{\sigma^2 L\Delta \log T}{T})^{1/4})$. Similar to the proof of Theorem 2.1, our construction is based on the "resisting oracle" argument, which we briefly sketch below. Without loss of generality, assume that AdaGrad is initialized with $x_1 = 0$. For some $\epsilon = \Omega(\sqrt{\frac{L\Delta \log T}{T}} + (\frac{\sigma^2 L\Delta \log T}{T})^{1/4})$, we aim to construct a function $p_{\Delta,L,\sigma,T}$ such that $p'_{\Delta,L,\sigma,T}(x_t) = -\epsilon$ for all $t \in [T]$ with the stochastic gradient oracle chosen as

$$\Pr(g_t = 0 \mid x_t) = \frac{\sigma^2}{\sigma^2 + \epsilon^2} \quad \text{and} \quad \Pr\Big(g_t = -\frac{\sigma^2 + \epsilon^2}{\epsilon} \mid x_t\Big) = \frac{\epsilon^2}{\sigma^2 + \epsilon^2}. \tag{9}$$

One can verify that $\mathbb{E}[g_t \mid x_t] = -\epsilon = p'(x_t)$ and $\mathbb{E}[|g_t - p'(x_t)|^2 \mid x_t] = \sigma^2$. Our key observation is that, under the stochastic gradient oracle in (9), the dynamic of AdaGrad can be modeled as a *random walk in one direction* and its query points can be determined in advance. Specifically, let $M_t$ denote the number of times the stochastic gradient is non-zero by time $t$. Since the non-zero stochastic gradients all take the same value, it follows from the update rule of AdaGrad that

$$\begin{cases} M_t = M_{t-1} + 1, \ x_{t+1} = x_t + \frac{\eta}{\sqrt{M_t}} & \text{if } g_t \ne 0 \text{ (with probability } \frac{\epsilon^2}{\sigma^2 + \epsilon^2}\text{)}; \\ M_t = M_{t-1}, \qquad x_{t+1} = x_t & \text{otherwise (with probability } \frac{\sigma^2}{\sigma^2 + \epsilon^2}\text{)}. \end{cases} \tag{10}$$

In particular, the points visited by AdaGrad belong to the set $\{\sum_{s=1}^{t} \frac{\eta}{\sqrt{s}} : t \ge 1\}$, which allows us to construct the function $p_{\Delta,L,\sigma,T}$.

Having defined the function $p_{\Delta,L,\sigma,T}$, we then set $f$ to be $f(\boldsymbol{x}) = \sum_{i=1}^{d} p_i(x^{(i)})$, where $p_i(\cdot) = p_{\Delta_i,L_i,\sigma_i,T}(\cdot)$ and $\sum_{i=1}^{d} \Delta_i = \Delta$. Thus, it follows that

$$\mathbb{E}\left[\min_{1 \le t \le T+1} \|\nabla f(\boldsymbol{x}_t)\|_1\right] = \Omega\Big(\sum_{i=1}^{d} \sqrt{\frac{L_i \Delta_i \log T}{T}} + \sum_{i=1}^{d} \Big(\frac{\sigma_i^2 L_i \Delta_i \log T}{T}\Big)^{\frac{1}{4}}\Big). \tag{11}$$

Finally, choosing $\Delta_i$ (for $i \in [d]$) properly to maximize the right-hand side of (11), we obtain the lower bound in Theorem 3.2. $\qquad \square$

Now let us compare our lower bound in Theorem 3.2 with the upper bound in (8), where we recall that $h(T)$ is a polynomial function of $T$ and problem parameters. We observe that the first noiseless term in our upper bound matches the corresponding term in our lower bound, up to an absolute constant. Notably, our lower bound shows that the additional logarithmic term in the upper bound is necessary, rather than being an artifact of the analysis. For the second noise-dependent term, the upper bound and the lower bound differ only in their dependence on $\boldsymbol{L}$ and $\boldsymbol{\sigma}$. Moreover, applying Hölder's inequality yields $(\sum_{i=1}^{d} \sigma_i^{2/3} L_i^{1/3})^3 \le \|\boldsymbol{\sigma}\|_1^2 \|\boldsymbol{L}\|_1$, and the equality holds when the noise variances and the Lipschitz parameters are aligned in a particular way. Hence, under certain conditions on $\boldsymbol{L}$ and $\boldsymbol{\sigma}$, the second terms also match up to an absolute constant. Finally, our upper bound contains an additional third term $\frac{\|\boldsymbol{\sigma}\|_1 \sqrt{\log h(T)}}{T^{\frac{1}{4}}}$, which is absent from our lower bound. It is an interesting open question whether this term can be improved.

The lower bound in Theorem 3.2 is specific to AdaGrad. In what follows, we present another lower bound that applies to all deterministic algorithms with access only to the first-order oracle, but only in the noiseless setting (where $\sigma_i = 0$ for all $i \in [d]$). This result is in the same spirit as Theorem 2.1, but here we use the $\ell_1$-norm of the gradient as the stationarity measure, as opposed to the $\ell_2$-norm. Since the proof technique is similar to the one in Theorem 2.1, we defer the proof to Appendix C.3.

**Theorem 3.3.** *Consider any deterministic algorithm $\mathcal{A}$ that only has access to the first-order oracle with an initial point $\boldsymbol{x}_1 \in \mathbb{R}^d$. For any positive vector $\mathbf{L} = [L_1, L_2, \ldots, L_d]$ and $\Delta_f > 0$, there exists a function $f : \mathbb{R}^d \to \mathbb{R}$ such that: (i) $f$ satisfies Assumption 2.4b and $f(\boldsymbol{x}_1) - \inf f \leq \Delta_f$; (ii) Algorithm $\mathcal{A}$ requires more than $\frac{\|\mathbf{L}\|_1 \Delta_f}{\epsilon^2}$ gradient queries to find a point $\hat{\mathbf{x}}$ with $\|\nabla f(\hat{\mathbf{x}})\|_1 < \epsilon$.*

Note that in the noiseless setting, our upper bound in (8) simplifies to $\mathcal{O}\left(\sqrt{\frac{\|\boldsymbol{L}\|_1 \Delta_F \log h(T)}{T}}\right)$, which is equivalent to $\tilde{\mathcal{O}}(\frac{\|\boldsymbol{L}\|_1 \Delta_F}{\epsilon^2})$ and matches the lower bound in Theorem 3.3, up to logarithmic terms.

## 4 $\ell_1$-NORM CONVERGENCE OF SGD: A LOWER BOUND

Having established the convergence of AdaGrad in terms of the gradient $\ell_1$-norm in the previous section, we now seek to compare it with the convergence rate of SGD. However, the existing convergence bounds for SGD use the $\ell_2$-norm of the gradient as the stationarity measure, making they are not directly comparable to our result in Theorem 3.1. To facilitate a rigorous comparison, our goal in this section is to provide a lower complexity bound for SGD with respect to the $\ell_1$-norm, which is shown in the following theorem.

**Theorem 4.1.** *Consider running SGD with update rule $\mathbf{x}_{t+1} = \mathbf{x}_t - \eta \boldsymbol{g}_t$ on a smooth function $f$ with a constant step size $\eta$. For any given positive vector $\mathbf{L} = [L_1, L_2, \ldots, L_d]$, non-negative vector $\boldsymbol{\sigma} = [\sigma_1, \sigma_2, \ldots, \sigma_d]$ and $\Delta_f > 0$, there exists a function $f : \mathbb{R}^d \to \mathbb{R}$ such that: (i) $f$ satisfies Assumption 2.4b and $f(\mathbf{x}_1) - \inf f \leq \Delta_f$; (ii) The stochastic gradient $\boldsymbol{g}_t$ satisfies Assumptions 2.2 and 2.3b; (iii) We have $\mathbb{E}\left[\min_{1 \leq t \leq T} \|\nabla f(\mathbf{x}_t)\|_1\right] = \Omega\left(\sqrt{\frac{d\|\boldsymbol{L}\|_\infty \Delta_f}{T}} + \frac{d^{1/4}\Delta_f^{1/4}(\sum_{i=1}^d \sigma_i \sqrt{L_i})^{1/2}}{T^{1/4}}\right)$ when $T$ is sufficiently large.*

*Proof Sketch.* We follow a similar approach as in Theorem 3.2. The function $f$ is constructed in the form of $f(\boldsymbol{x}) = \sum_{i=1}^d p_i(x^{(i)})$, where $x^{(i)}$ denotes the $i$-th coordinate of the vector $\boldsymbol{x} \in \mathbb{R}^d$ and $p_i : \mathbb{R} \to \mathbb{R}$ is a one-dimensional function to be determined. Similar to AdaGrad, our key observation is that running SGD on $f$ is equivalent to running SGD with the same step size $\eta$ for each of the one-dimensional function $p_i$ in parallel, and thus it is sufficient to characterize the complexity lower bound in the one-dimensional setting.

Extending the construction in (Abbaszadehpeivasti et al., 2022, Proposition 4) to the stochastic setting, we show that there exists a one-dimensional function $p_{\Delta, L, \sigma, \eta, T}(\cdot)$ and an associated stochastic oracle such that: (i) Its gradient is $L$-Lipschitz and the initial function value gap is bounded by $\Delta$; (ii) The stochastic gradient oracle is unbiased with bounded variance $\sigma^2$; (iii) The iterates of SGD with step size $\eta$ satisfy $\mathbb{E}\left[\min_{1 \leq t \leq T} |p'(x_t)|\right] \geq \sqrt{2L\Delta}$ if $\eta \geq \frac{2}{L}$, and $\mathbb{E}\left[\min_{1 \leq t \leq T} |p'(x_t)|\right] \geq \max\left\{\frac{1}{2}\sqrt{\frac{\Delta}{2\eta T + \frac{1}{2L}}}, \min\left\{\sigma\sqrt{\frac{L\eta}{2}}, \sqrt{2L\Delta}\right\}\right\}$ otherwise. Given this result, we then set $f(\boldsymbol{x}) = \sum_{i=1}^d p_{\frac{\Delta}{d}, L_i, \sigma_i, T, \eta}(x^{(i)})$, where $x^{(i)}$ denotes the $i$-th coordinate of $\boldsymbol{x}$. By considering different choices of the step size $\eta$ and establishing a lower bound in each case, we arrive at the final result. $\qquad\square$

From Theorem 4.1, we observe that the convergence rate of SGD exhibits a similar dependence on the number of iterations $T$ as AdaGrad. However, a key distinction lies in the explicit dependence on the dimension $d$. In the next section, we provide a detailed comparison between the lower bound of SGD with the upper bound of AdaGrad.

## 5 COMPARISON BETWEEN ADAGRAD AND SGD

In this section, we compare the rate obtained in Theorem 3.1 for AdaGrad with the convergence lower bound of SGD in Theorem 4.1. Inspired by the analysis in Bernstein et al. (2018), we introduce two density functions for this comparison. We define the density functions $\phi : \mathbb{R}^d \to [0, 1]$ as follows:

$$\phi(\boldsymbol{v}) := \frac{\|\boldsymbol{v}\|_1^2}{d\|\boldsymbol{v}\|_2^2} \in \left[\frac{1}{d}, 1\right] \quad \text{and} \quad \tilde{\phi}(\boldsymbol{v}) := \frac{\|\boldsymbol{v}\|_1}{d\|\boldsymbol{v}\|_\infty} \in \left[\frac{1}{d}, 1\right]. \tag{12}$$

Specifically, a larger value of $\phi(\boldsymbol{v})$ or $\tilde{\phi}(\boldsymbol{v})$ indicates that the vector $\boldsymbol{v}$ is denser. Using this notation, we can write $\|\boldsymbol{\sigma}_2\|_2^2 = \frac{\|\boldsymbol{\sigma}\|_1^2}{d\phi(\boldsymbol{\sigma})}$ and $\|\boldsymbol{L}\|_\infty = \frac{\|\boldsymbol{L}\|_1}{d\tilde{\phi}(\boldsymbol{L})}$, and the lower bound in Theorem 4.1 for SGD becomes

$$\min_{t=1,\ldots,T} \mathbb{E}\left[\|\nabla F(\boldsymbol{w}_t)\|_1\right] = \Omega\left(\sqrt{\frac{\|\boldsymbol{L}\|_1 \Delta_F}{\tilde{\phi}(\boldsymbol{L})T}} + \left(\frac{R^2 \|\boldsymbol{\sigma}\|_1^2 \|\boldsymbol{L}\|_1 \Delta_F}{\phi(\boldsymbol{\sigma})T}\right)^{\frac{1}{4}}\right), \quad (13)$$

where

$$R = \frac{\sum_{i=1}^d \sigma_i \sqrt{L_i}}{\|\boldsymbol{\sigma}\|_2 \sqrt{\|\boldsymbol{L}\|_1}} \in [0,1] \quad (14)$$

is the cosine similarity between the two vectors $[\sigma_1,\ldots,\sigma_d] \in \mathbb{R}^d$ and $[\sqrt{L_1},\ldots,\sqrt{L_d}] \in \mathbb{R}^d$. To facilitate the comparison, we first translate the convergence rates of AdaGrad in (8) and SGD in (13) into equivalent iteration complexity bounds. Specifically, to find an $\epsilon$-stationary point in terms of the $\ell_1$-norm, we observe that the required number of iterations is

$$\tilde{\mathcal{O}}\left(\frac{\|\boldsymbol{L}\|_1 \Delta_F}{\epsilon^2} + \frac{\|\boldsymbol{\sigma}\|_1^2 \|\boldsymbol{L}\|_1 \Delta_F}{\epsilon^4} + \frac{\|\boldsymbol{\sigma}\|_1^4}{\epsilon^4}\right) \quad \text{for AdaGrad}, \quad (15)$$

$$\text{and} \quad \Omega\left(\frac{\|\boldsymbol{L}\|_1 \Delta_F}{\tilde{\phi}(\boldsymbol{L})\epsilon^2} + \frac{R^2 \|\boldsymbol{\sigma}\|_1^2 \|\boldsymbol{L}\|_1 \Delta_F}{\phi(\boldsymbol{\sigma})\epsilon^4}\right) \quad \text{for SGD}. \quad (16)$$

Except for the additional term $\frac{\|\boldsymbol{\sigma}\|_1^4}{\epsilon^4}$ in (15), we observe that the two bounds in (15) and (16) are similar. If we assume that the noise is relatively small, i.e., $\|\boldsymbol{\sigma}\|_1 \ll \sqrt{\|\boldsymbol{L}\|_1 \Delta_F}$, the first two terms dominate. We can make the following observations:

- Since $\tilde{\phi}(\boldsymbol{L}) \in [\frac{1}{d}, 1]$, for the first noiseless term in (15) and (16), AdaGrad is never worse than SGD and outperforms SGD by a factor of $\tilde{\phi}(\boldsymbol{L})$. In particular, in the extreme case where $\tilde{\phi}(\boldsymbol{L}) = \frac{1}{d}$, i.e., the vector $\boldsymbol{L}$ is sparse, AdaGrad reduces the bound of SGD by a factor of $d$.
- Since $R \in [0,1]$ and $\phi(\boldsymbol{\sigma}) \in [\frac{1}{d}, 1]$, the second noise-dependent term in AdaGrad can be either improve or worsen compared to SGD. In the extreme case where $R = 1$ and $\phi(\boldsymbol{\sigma}) = \frac{1}{d}$, i.e., the two vectors $[\sigma_1,\ldots,\sigma_d]$ and $[\sqrt{L_1},\ldots,\sqrt{L_d}]$ are aligned and the vector $\boldsymbol{\sigma}$ is sparse, then AdaGrad similarly reduces the bound of SGD by a factor of $d$.

To our knowledge, our results provide the first problem setting where AdaGrad provably achieves a better dimensional dependence than SGD in the non-convex setting. We note that our discussions here mirror the comparison between AdaGrad and Online Gradient Descent in (McMahan & Streeter, 2010; Duchi et al., 2011) regarding online convex optimization problems. Similarly, depending on the geometry of the feasible set and the density of the gradient vectors, it is shown that the rate of AdaGrad can be better or worse by a factor of $\sqrt{d}$. In this sense, our result complements this classical result and demonstrates that a similar phenomenon also occurs in the non-convex setting.

# 6 CONCLUSION

In this paper, we provided a theoretical justification for the advantage of AdaGrad over SGD in stochastic non-convex optimization. We first discussed the impossibility of showing any convergence rate improvement over SGD under the standard assumptions of Lipschitz gradients and bounded variance, as well as using the gradient's $\ell_2$-norm as the stationarity measure. Motivated by this observation, we introduced two refined assumptions on the Lipschitz constants and gradient noise of the objective (Assumptions 2.3b and 2.4b) and proposed using the gradient $\ell_1$-norm as the stationarity measure, which better suit the coordinate-wise nature of adaptive gradient methods. Under these refined assumptions, We established a convergence rate for AdaGrad (Theorem 3.1) and a complexity lower bound for SGD (Theorem 4.1) in terms of the gradient's $\ell_1$-norm. Notably, by comparing AdaGrad's *upper bound* with SGD's *lower bound*, we demonstrated that the complexity of AdaGrad can be better than that of SGD by a factor of $d$. To our knowledge, this is the first result showing a provable advantage of adaptive gradient methods over SGD in non-convex optimization. In addition, by presenting two lower bounds, we established that the noiseless term in our upper bound for AdaGrad is unimprovable up to a logarithmic factor (Theorems 3.2 and 3.3).

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

APPENDIX

# A  COMPARISON WITH EXISTING RESULTS ON ADAGRAD

Most of the existing works use the $\ell_2$-norm as a measure of convergence (Shen et al., 2023; Défossez et al., 2022; Wang et al., 2023; Hong & Lin, 2024; Zhou et al., 2024). The state-of-the-art result is Zhou et al. (2024): with a fine-tuned step size, the authors show that, with high probability, AdaGrad satisfies $\frac{1}{T}\sum_{t=1}^{T}\|\nabla F(\boldsymbol{w}_t)\|_2^2 = \mathcal{O}\left(\frac{d}{T} + \frac{d^{1/2}}{T^{1/2}}\right)$. If we use this result to show a bound for the $\ell_1$-norm, since $\|\nabla F(\boldsymbol{w}_t)\|_1 = \Theta(\sqrt{d}\|\nabla F(\boldsymbol{w}_t)\|_2)$ in the worst case, the upper bound becomes $\min_{t=1,\ldots,T}\|\nabla F(\boldsymbol{w}_t)\|_1 = \mathcal{O}\left(\frac{d}{\sqrt{T}} + \frac{d^{3/4}}{T^{1/4}}\right)$, which is worse than our bound by at least a factor of $d^{1/4}$.

Also, in Liu et al. (2023), the authors considered the case that that the function is $L$-smooth and the noise of gradient is coordinate-wise subgaussian, i.e., $\mathbb{E}\left[\exp(\lambda^2(g_{t,i} - \nabla_i F(\boldsymbol{w}_t))^2)\right] \leq \exp(\lambda^2\sigma_i^2)$ for all $\lambda$ such that $|\lambda| < \frac{1}{\sigma_i}$. Note that the subgaussian noise assumption is stronger than the bounded variance assumption in Assumption 2.3b. Under these assumptions, they characterized the convergence rate of AdaGrad in terms of the averaged $\ell_1$-norm of the gradient and their result is no better than $\tilde{\mathcal{O}}\left(\frac{\Delta_1}{\sqrt{T}} + \frac{dL}{\sqrt{T}} + \frac{\sqrt{\Delta_F\|\boldsymbol{\sigma}\|_1}}{T^{1/4}} + \frac{\sqrt{d}\|\boldsymbol{\sigma}\|_1}{T^{1/4}} + \frac{\sqrt{dL\|\boldsymbol{\sigma}\|_1}}{T^{1/4}}\right)$. Compared to our bounds in (8), we observe that their term $\frac{\sqrt{d}\|\boldsymbol{\sigma}\|_1}{T^{1/4}}$ is worse than the corresponding term in ours by a factor of $\sqrt{d}$. Moreover, in the worst case where $\frac{\|\boldsymbol{L}\|_1}{\|\boldsymbol{L}\|_\infty} = \Theta(d)$ and $\frac{\|\boldsymbol{\sigma}\|_1}{\|\boldsymbol{\sigma}\|_2} = \Theta(\sqrt{d})$, their overall bound is worse than ours by a factor of $\sqrt{d}$.

# B  PROOF OF THEOREM 3.1

In this section, we prove Theorem 3.1. Recall that we define $\eta_{t,i} = \frac{\eta}{b_{t,i}+\delta}$ and thus AdaGrad can be rewritten as $w_{t+1,i} = w_{t,i} - \eta_{t,i}g_{t,i}$ for $i \in [d]$. Our starting point is applying Assumption 2.4b to $\boldsymbol{w}_t$ and $\boldsymbol{w}_{t+1}$, yielding:

$$F(\boldsymbol{w}_{t+1}) \leq F(\boldsymbol{w}_t) + \langle\nabla F(\boldsymbol{w}_t), \boldsymbol{w}_{t+1} - \boldsymbol{w}_t\rangle + \sum_{i=1}^{d}\frac{L_i}{2}|w_{t+1,i} - w_{t,i}|^2$$

$$= F(\boldsymbol{w}_t) - \sum_{i=1}^{d}\eta_{t,i}\nabla_i F(\boldsymbol{w}_t)g_{t,i} + \sum_{i=1}^{d}\frac{L_i}{2}\eta_{t,i}^2 g_{t,i}^2. \tag{17}$$

If the step size $\eta_{t,i}$ were conditionally independent of the stochastic gradient $g_{t,i}$, then by taking the conditional expectation with respect to $\mathcal{F}_{t-1}$, the second term in the right-hand side of (17) would result in $-\eta_{t,i}\nabla_i F(\boldsymbol{w}_t)\mathbb{E}\left[g_{t,i} \mid \mathcal{F}_{t-1}\right] = -\eta_{t,i}\nabla_i F(\boldsymbol{w}_t)^2$ by Assumption 2.2. However, as mentioned in the proof sketch, the difficulty is that the step size $\eta_{t,i}$ is computed using the stochastic gradient at the current iterate $\boldsymbol{w}_t$, and consequently $\mathbb{E}\left[\eta_{t,i}g_{t,i} \mid \mathcal{F}_{t-1}\right] \neq \eta_{t,i}\mathbb{E}\left[g_{t,i} \mid \mathcal{F}_{t-1}\right]$ in general.

Following Ward et al. (2020); Faw et al. (2022), we tackle this challenge by introducing the decorrelated step size $\hat{\eta}_{t,i}$ in (2), which serves as a "proxy" of the step size that is decorrelated from $\boldsymbol{g}_t$. Specifically, note that $\hat{\eta}_{t,i}$ belongs to the filtration $\mathcal{F}_{t-1}$ and thus $\mathbb{E}\left[\hat{\eta}_{t,i}\nabla_i F(\boldsymbol{w}_t)g_{t,i} \mid \mathcal{F}_{t-1}\right] = \hat{\eta}_{t,i}\nabla_i F(\boldsymbol{w}_t)^2$, leading to the desired squared gradient that we aim to bound. Equipped with the decorrelated step size, in the following lemma we prove an upper bound on a (weighted) gradient square norm at the current iterate $\boldsymbol{w}_t$.

**Lemma B.1.** *Suppose Assumptions 2.2 and 2.4b hold. Consider the update rule in AdaGrad and recall the decorrelated step sizes defined in (2). Then we have*

$$\sum_{i=1}^{d}\hat{\eta}_{t,i}\nabla_i F(\boldsymbol{w}_t)^2 \leq F(\boldsymbol{w}_t) - \mathbb{E}\left[F(\boldsymbol{w}_{t+1}) \mid \mathcal{F}_{t-1}\right] + \sum_{i=1}^{d}\mathbb{E}\left[(\hat{\eta}_{t,i} - \eta_{t,i})\nabla_i F(\boldsymbol{w}_t)g_{t,i} \mid \mathcal{F}_{t-1}\right]$$

$$+ \sum_{i=1}^{d}\frac{L_i}{2}\mathbb{E}\left[\eta_{t,i}^2 g_{t,i}^2 \mid \mathcal{F}_{t-1}\right]. \tag{18}$$

*Proof.* Taking the expectation with respect to $\mathcal{F}_{t-1}$ in (17), we obtain:

$$\mathbb{E}\left[F(\boldsymbol{w}_{t+1}) \mid \mathcal{F}_{t-1}\right] - F(\boldsymbol{w}_t) = -\sum_{i=1}^{d}\left(\mathbb{E}\left[\eta_{t,i}\nabla_i F(\boldsymbol{w}_t)g_{t,i} \mid \mathcal{F}_{t-1}\right] + \frac{L_i}{2}\mathbb{E}\left[\eta_{t,i}^2 g_{t,i}^2 \mid \mathcal{F}_{t-1}\right]\right). \quad (19)$$

Since $\hat{\eta}_{t,i}$ is independent from $g_{t,i}$ conditioned on $\mathcal{F}_{t-1}$, it follows from Assumption 2.2 that $\mathbb{E}\left[\hat{\eta}_{t,i}\nabla_i F(\boldsymbol{w}_t)g_{t,i} \mid \mathcal{F}_{t-1}\right] = \hat{\eta}_{t,i}\nabla_i F(\boldsymbol{w}_t)\mathbb{E}\left[g_{t,i} \mid \mathcal{F}_{t-1}\right] = \hat{\eta}_{t,i}\nabla_i F(\boldsymbol{w}_t)^2$. Hence, we get

$$\mathbb{E}\left[\eta_{t,i}\nabla_i F(\boldsymbol{w}_t)g_{t,i} \mid \mathcal{F}_{t-1}\right] = \mathbb{E}\left[\hat{\eta}_{t,i}\nabla_i F(\boldsymbol{w}_t)g_{t,i} \mid \mathcal{F}_{t-1}\right] + \mathbb{E}\left[(\eta_{t,i} - \hat{\eta}_{t,i})\nabla_i F(\boldsymbol{w}_t)g_{t,i} \mid \mathcal{F}_{t-1}\right]$$

$$= \hat{\eta}_{t,i}\nabla_i F(\boldsymbol{w}_t)^2 + \mathbb{E}\left[(\eta_{t,i} - \hat{\eta}_{t,i})\nabla_i F(\boldsymbol{w}_t)g_{t,i} \mid \mathcal{F}_{t-1}\right].$$

Combining this with (19), this further implies that

$$\mathbb{E}\left[F(\boldsymbol{w}_{t+1}) \mid \mathcal{F}_{t-1}\right] - F(\boldsymbol{w}_t) \leq \sum_{i=1}^{d}\Big(-\hat{\eta}_{t,i}\nabla_i F(\boldsymbol{w}_t)^2 - \mathbb{E}\left[(\eta_{t,i} - \hat{\eta}_{t,i})\nabla_i F(\boldsymbol{w}_t)g_{t,i} \mid \mathcal{F}_{t-1}\right]$$

$$+ \frac{L_i}{2}\mathbb{E}\left[\eta_{t,i}^2 g_{t,i}^2 \mid \mathcal{F}_{t-1}\right]\Big).$$

Rearranging the above inequality leads to (18). $\qquad\square$

In Lemma B.1, the left-hand side is a weighted version of the squared gradient norm at $\boldsymbol{w}_t$, where the weights for each coordinate are given by the decorrelated step sizes $\hat{\eta}_{t,i}$. Note that this is the key difference compared to the analysis of AdaGrad-Norm in Faw et al. (2022). Indeed, for AdaGrad-Norm, the left-hand side will become $\hat{\eta}_t\|\nabla F(\mathbf{w}_t)\|^2$, and thus the squared $\ell_2$-norm of the gradient naturally arises from the analysis. On the other hand, as we shall see later, in our case $\ell_2$-norm is not the best choice of the norm and instead we will relate the left-hand side in (18) to the $\ell_1$-norm of the gradient.

In light of Lemma B.1, we need to manage the *bias term* $\sum_{i=1}^{d}\mathbb{E}\left[(\hat{\eta}_{t,i} - \eta_{t,i})\nabla_i F(\boldsymbol{w}_t)g_{t,i} \mid \mathcal{F}_{t-1}\right]$, which is due to the difference between the step size $\eta_{t,i}$ and its decorrelated version $\hat{\eta}_{t,i}$, and a *quadratic term* $\sum_{i=1}^{d}\mathbb{E}[\eta_{t,i}^2 g_{t,i}^2]$, which comes from Assumption 2.4b. The following lemma addresses these two terms and the proofs for these two results are presented in Appendix B.1.

**Lemma B.2.** *Consider the update rule in AdaGrad. For any $t \in [T]$ and $i \in [d]$, we have*

$$\mathbb{E}\left[(\hat{\eta}_{t,i} - \eta_{t,i})\nabla_i F(\boldsymbol{w}_t)g_{t,i} \mid \mathcal{F}_{t-1}\right] \leq \frac{\hat{\eta}_{t,i}}{2}\nabla_i F(\boldsymbol{w}_t)^2 + \frac{2\sigma_i}{\eta}\mathbb{E}\left[\eta_{t,i}^2 g_{t,i}^2 \mid \mathcal{F}_{t-1}\right]. \quad (20)$$

*Moreover, we have*

$$\mathbb{E}\left[\sum_{t=1}^{T}\eta_{t,i}^2 g_{t,i}^2\right] \leq \eta^2 \log h(T), \quad (21)$$

*where* $h(T) = 1 + \frac{T\|\boldsymbol{\sigma}\|_\infty^2}{\delta^2} + \frac{T(\|\nabla F(\boldsymbol{w}_1)\|_\infty + \eta\sqrt{\|\boldsymbol{L}\|_\infty\|\boldsymbol{L}\|_1 T})^2}{\delta^2}$.

The first result in Lemma B.2 shows that for each coordinate $i \in [d]$, we can upper bound the bias term in terms of the squared gradient $\frac{\hat{\eta}_{t,i}}{2}\nabla_i F(\boldsymbol{w}_t)^2$ and the quadratic term $\mathbb{E}\left[\eta_{t,i}^2 g_{t,i}^2\right]$. The second result in the above lemma shows that the accumulation of the quadratic terms $\eta_{t,i}^2 g_{t,i}^2$ over $T$ iterations can be bounded in expectation by $\mathcal{O}(\eta^2 \log(T/\delta))$. By combining Lemma B.2 with Lemma B.1, we obtain the following key corollary.

**Corollary B.3.** *Recall the definition of $h(T)$ in Lemma B.2. For AdaGrad, we have*

$$\mathbb{E}\left[\sum_{t=1}^{T}\sum_{i=1}^{d}\frac{\hat{\eta}_{t,i}}{2}\nabla_i F(\boldsymbol{w}_t)^2\right] \leq F(\boldsymbol{w}_1) - F^* + \left(2\eta\|\boldsymbol{\sigma}\|_1 + \frac{\eta^2\|\boldsymbol{L}\|_1}{2}\right)\log h(T). \quad (22)$$

*Proof.* By applying (20) to (18) in Lemma B.1, we obtain that

$$\sum_{i=1}^{d}\hat{\eta}_{t,i}\nabla_i F(\boldsymbol{w}_t)^2 \leq F(\boldsymbol{w}_t) - \mathbb{E}\left[F(\boldsymbol{w}_{t+1}) \mid \mathcal{F}_{t-1}\right] + \sum_{i=1}^{d}\frac{\hat{\eta}_{t,i}}{2}\nabla_i F(\boldsymbol{w}_t)^2$$

$$+ \sum_{i=1}^{d}\left(\frac{L_i}{2} + \frac{2\sigma_i}{\eta}\right)\mathbb{E}\left[\eta_{t,i}^2 g_{t,i}^2 \mid \mathcal{F}_{t-1}\right].$$

By merging terms and taking the expectation of both sides of the inequality, we further have

$$\mathbb{E}\left[\sum_{i=1}^{d}\frac{\hat{\eta}_{t,i}}{2}\nabla_i F(\boldsymbol{w}_t)^2\right] \leq \mathbb{E}\left[F(\boldsymbol{w}_t) - F(\boldsymbol{w}_{t+1})\right] + \sum_{i=1}^{d}\left(\frac{\eta^2 L_i}{2} + 2\eta\sigma_i\right)\mathbb{E}\left[\eta_{t,i}^2 g_{t,i}^2\right].$$

Now we sum the above the inequality over $t = 1, \ldots, T$ to get

$$\mathbb{E}\left[\sum_{t=1}^{T}\sum_{i=1}^{d}\frac{\hat{\eta}_{t,i}}{2}\nabla_i F(\boldsymbol{w}_t)^2\right] \leq F(\boldsymbol{w}_1) - \mathbb{E}\left[F(\boldsymbol{w}_{T+1})\right] + \sum_{i=1}^{d}\left(2\eta\sigma_i + \frac{L_i\eta^2}{2}\right)\mathbb{E}\left[\sum_{t=1}^{T}\eta_{t,i}^2 g_{t,i}^2\right]$$

$$\leq F(\boldsymbol{w}_1) - F^* + \sum_{i=1}^{d}\left(2\eta\sigma_i + \frac{L_i\eta^2}{2}\right)\log h(T)$$

$$= F(\boldsymbol{w}_1) - F^* + \left(2\eta\|\boldsymbol{\sigma}\|_1 + \frac{\|\boldsymbol{L}\|_1\eta^2}{2}\right)\log h(T),$$

where we used Assumption 2.1 and (21) in the second inequality. This completes the proof. $\qquad\square$

To simplify the notation, let us denote the right-hand side of (22) by $Q$. This implies that, if we ignore the logarithmic term, we have $Q = \tilde{\mathcal{O}}\left(F(\boldsymbol{w}_1) - F^* + \eta\|\boldsymbol{\sigma}\|_1 + \eta^2\|\boldsymbol{L}\|_1\right)$. Corollary B.3 shows that the sum of weighted squared gradient norms is bounded by a constant depending on problem parameters, up to log factors. Hence, the remaining task is to establish lower bounds on the step sizes $\hat{\eta}_{t,i}$. For instance, if we were able to show that all the step sizes $\hat{\eta}_{t,i}$ are uniformly lower bounded by $\tilde{\Omega}(\frac{1}{\sqrt{T}})$, then Corollary B.3 would immediately imply a rate of $\tilde{O}(\frac{1}{T^{1/4}})$ in terms of the gradient $\ell_2$-norm $\|\nabla F(\boldsymbol{w}_t)\|_2$. However, there are several challenges: (i) The step sizes $\hat{\eta}_{t,i}$ are determined by the observed stochastic gradient rather than specified by the user. (ii) To further complicate the issue, due to correlation between the step size $\hat{\eta}_{t,i}$ and the iterate $\boldsymbol{w}_t$, this implies that $\mathbb{E}\left[\hat{\eta}_{t,i}\nabla_i F(\boldsymbol{w}_t)^2\right] \neq \mathbb{E}\left[\hat{\eta}_{t,i}\right]\mathbb{E}\left[\nabla_i F(\boldsymbol{w}_t)^2\right]$ and hence a lower bound on $\mathbb{E}\left[\hat{\eta}_{t,i}\right]$ would not suffice. (iii) Finally, since the step sizes for each coordinate are updated independently, it is unclear how to construct a uniform lower bound across all the coordinates.

As mentioned in the proof sketch, to address the last challenge, we study each coordinate and construct a uniform lower bound on $\hat{\eta}_{t,i}$ for $t \in [T]$. Specifically, for each coordinate $i \in [d]$, we define a new auxiliary step size $\tilde{\eta}_{T,i}$ as in (4). From (2) and $b_{t-1,i} = \sum_{s=1}^{t-1} g_{s,i}^2$ in (AdaGrad), we have $\hat{\eta}_{t,i} \geq \tilde{\eta}_{T,i}$ for all $t \in [T]$. To address the second issue, we separate the step sizes from the gradients as follows:

$$\mathbb{E}\left[\sum_{t=1}^{T}\frac{\hat{\eta}_{t,i}}{2}\nabla_i F(\boldsymbol{w}_t)^2\right] \geq \mathbb{E}\left[\frac{\tilde{\eta}_{T,i}}{2}\sum_{t=1}^{T}\nabla_i F(\boldsymbol{w}_t)^2\right] \geq \frac{\mathbb{E}\left[\sqrt{\sum_{t=1}^{T}\nabla_i F(\boldsymbol{w}_t)^2}\right]^2}{\mathbb{E}\left[\frac{2}{\tilde{\eta}_{T,i}}\right]}, \quad (23)$$

where we used the elementary inequality that $\mathbb{E}\left[\frac{X^2}{Y}\right] \geq \frac{\mathbb{E}[X]^2}{\mathbb{E}[Y]}$ for any two positive random variables $X$ and $Y$. Hence, in the following lemma, we will establish an upper bound on $\mathbb{E}\left[\frac{1}{\tilde{\eta}_{T,i}}\right]$, instead of directly lower bounding $\mathbb{E}\left[\tilde{\eta}_{T,i}\right]$.

**Lemma B.4.** *Consider the step size $\tilde{\eta}_{T,i}$ defined in (4). For any $i \in [d]$, we have*

$$\mathbb{E}\left[\frac{1}{\tilde{\eta}_{T,i}}\right] \leq \frac{\sigma_i\sqrt{2T} + \delta}{\eta} + \frac{\sqrt{3}}{\eta}\mathbb{E}\left[\sqrt{\sum_{t=1}^{T}\nabla_i F(\boldsymbol{w}_t)^2}\right].$$

*Proof.* From the definition of $\tilde{\eta}_{T,i}$ and using $b_{t-1,i}^2 = \sum_{s=1}^{t-1} g_{s,i}^2 \leq \sum_{t=1}^{T-1} g_{t,i}^2$, we have

$$\mathbb{E}\left[\frac{\eta}{\tilde{\eta}_{T,i}}\right] \leq \mathbb{E}\left[\sqrt{\sum_{t=1}^{T}g_{t,i}^2 + \sigma_i^2 + \sum_{t=1}^{T}\nabla_i F(\boldsymbol{w}_t)^2 + \delta}\right].$$

We then can use the upper bound of $g_{t,i}^2 \leq 2((g_{t,i} - \nabla_i F(\boldsymbol{w}_t))^2 + \nabla_i F(\boldsymbol{w}_t)^2)$:

$$\mathbb{E}\left[\frac{\eta}{\tilde{\eta}_{T,i}}\right] \leq \mathbb{E}\left[\sqrt{\sum_{t=1}^{T} 2((g_{t,i} - \nabla_i F(\boldsymbol{w}_t))^2 + \nabla_i F(\boldsymbol{w}_t)^2) + \sigma_i^2 + \sum_{t=1}^{T} \nabla_i F(\boldsymbol{w}_t)^2 + \delta}\right]$$

$$= \mathbb{E}\left[\sqrt{2\sum_{t=1}^{T-1}(g_{t,i} - \nabla_i F(\boldsymbol{w}_t))^2 + 3\sum_{t=1}^{T} \nabla_i F(\boldsymbol{w}_t)^2 + \sigma_i^2 + \delta}\right]$$

$$\leq \mathbb{E}\left[\sqrt{2\sum_{t=1}^{T-1}(g_{t,i} - \nabla_i F(\boldsymbol{w}_t))^2 + \sigma_i^2}\right] + \mathbb{E}\left[\sqrt{3\sum_{t=1}^{T} \nabla_i F(\boldsymbol{w}_t)^2}\right] + \delta.$$

Applying Jensen's inequality and the bounded variance from Assumption 2.3b, we get

$$\mathbb{E}\left[\frac{\eta}{\tilde{\eta}_{T,i}}\right] \leq \sqrt{2\sum_{t=1}^{T-1}\mathbb{E}\left[(g_{t,i} - \nabla_i F(\boldsymbol{w}_t))^2\right] + \sigma_i^2} + \mathbb{E}\left[\sqrt{3\sum_{t=1}^{T} \nabla_i F(\boldsymbol{w}_t)^2}\right] + \delta$$

$$\leq \sqrt{2T\sigma_i^2} + \sqrt{3}\mathbb{E}\left[\sqrt{\sum_{t=1}^{T} \nabla_i F(\boldsymbol{w}_t)^2}\right] + \delta$$

Rearranging the terms immediately leads to the stated lemma. $\qquad\square$

Lemma B.4 establishes an upper bound on $\mathbb{E}\left[\frac{1}{\tilde{\eta}_{T,i}}\right]$ in terms of the sum $\mathbb{E}\left[\sqrt{\sum_{t=1}^{T} \nabla_i F(\boldsymbol{w}_t)^2}\right]$, which also appears on the right hand side of (5). By combining Corollary B.3, (5) and Lemma B.4, we arrive at the following lemma.

**Lemma B.5.** *Consider the update in AdaGrad and recall that $Q$ denotes the right-hand side in (3). It holds that*

$$\mathbb{E}\left[\sum_{i=1}^{d}\sqrt{\sum_{t=1}^{T} \nabla_i F(\boldsymbol{w}_t)^2}\right] \leq \frac{2\sqrt{3}}{\eta}Q + \sqrt{\frac{2d\delta Q}{\eta}} + 2\sqrt{\frac{\|\boldsymbol{\sigma}\|_1 Q}{\eta}}T^{\frac{1}{4}}. \tag{24}$$

*Proof.* It follows from (23) that

$$\mathbb{E}\left[\sqrt{\sum_{t=1}^{T} \nabla_i F(\boldsymbol{w}_t)^2}\right]^2 \leq \mathbb{E}\left[\sum_{t=1}^{T}\frac{\hat{\eta}_{t,i}}{2}\nabla_i F(\boldsymbol{w}_t)^2\right]\mathbb{E}\left[\frac{2}{\tilde{\eta}_{T,i}}\right].$$

Using the result from Lemma B.4, we get a quadratic inequality as follows:

$$\mathbb{E}\left[\sqrt{\sum_{t=1}^{T} \nabla_i F(\boldsymbol{w}_t)^2}\right] \leq \sqrt{\mathbb{E}\left[\sum_{t=1}^{T}\frac{\hat{\eta}_{t,i}}{2}\nabla_i F(\boldsymbol{w}_t)^2\right]}\sqrt{\mathbb{E}\left[\frac{2}{\tilde{\eta}_{T,i}}\right]}$$

$$\leq \sqrt{\frac{2}{\eta}}\sqrt{\mathbb{E}\left[\sum_{t=1}^{T}\frac{\hat{\eta}_{t,i}}{2}\nabla_i F(\boldsymbol{w}_t)^2\right]}\sqrt{(\sigma_i\sqrt{2T} + \delta) + \sqrt{3}\mathbb{E}\left[\sqrt{\sum_{t=1}^{T} \nabla_i F(\boldsymbol{w}_t)^2}\right]}$$

Solving the quadratic we have he following bound,

$$\mathbb{E}\left[\sqrt{\sum_{t=1}^{T} \nabla_i F(\boldsymbol{w}_t)^2}\right] \leq \frac{2\sqrt{3}}{\eta}\mathbb{E}\left[\sum_{t=1}^{T}\frac{\hat{\eta}_{t,i}}{2}\nabla_i F(\boldsymbol{w}_t)^2\right] + \sqrt{\frac{2}{\eta}}\sqrt{(\sigma_i\sqrt{2T} + \delta)}\sqrt{\mathbb{E}\left[\sum_{t=1}^{T}\frac{\hat{\eta}_{t,i}}{2}\nabla_i F(\boldsymbol{w}_t)^2\right]}$$

Combining the bounds from all the coordinates and using the Cauchy-Schwartz inequality for the second term:

$$\mathbb{E}\left[\sum_{i=1}^{d}\sqrt{\sum_{t=1}^{T}\nabla_i F(\boldsymbol{w}_t)^2}\right] \leq \frac{2\sqrt{3}}{\eta}\mathbb{E}\left[\sum_{i=1}^{d}\sum_{t=1}^{T}\frac{\hat{\eta}_{t,i}}{2}\nabla_i F(\boldsymbol{w}_t)^2\right]$$

$$+ \sqrt{\frac{2}{\eta}}\sqrt{\sum_{i=1}^{d}\sigma_i\sqrt{2T} + d\delta}\sqrt{\mathbb{E}\left[\sum_{i=1}^{d}\sum_{t=1}^{T}\frac{\hat{\eta}_{t,i}}{2}\nabla_i F(\boldsymbol{w}_t)^2\right]} \quad (25)$$

We can further bound the term using the result from Corollary B.3,

$$\mathbb{E}\left[\sum_{i=1}^{d}\sqrt{\sum_{t=1}^{T}\nabla_i F(\boldsymbol{w}_t)^2}\right] \leq \frac{2\sqrt{3}}{\eta}\left(F(\boldsymbol{w}_1) - F^* + \left(2\eta\|\boldsymbol{\sigma}\|_1 + \frac{\eta^2\|\boldsymbol{L}\|_1}{2}\right)\log h(T)\right)$$

$$+ \sqrt{\frac{2}{\eta T}}\sqrt{(\|\boldsymbol{\sigma}\|_1\sqrt{2T} + d\delta)}\sqrt{F(\boldsymbol{w}_1) - F^* + \left(2\eta\|\boldsymbol{\sigma}\|_1 + \frac{\eta^2\|\boldsymbol{L}\|_1}{2}\right)\log h(T)}$$

where $h(T)$ is defined in Lemma B.2. This completes the proof. $\square$

Finally, we relate the left-hand side of (24) to the $\ell_1$-norm of the gradients. Specifically, we can write:

$$\frac{1}{T}\sum_{t=1}^{T}\|\nabla F(\boldsymbol{w}_t)\|_1 = \frac{1}{T}\sum_{t=1}^{T}\sum_{i=1}^{d}|\nabla_i F(\boldsymbol{w}_t)| = \frac{1}{T}\sum_{i=1}^{d}\sum_{t=1}^{T}|\nabla_i F(\boldsymbol{w}_t)| \leq \frac{1}{\sqrt{T}}\sum_{i=1}^{d}\sqrt{\sum_{t=1}^{T}|\nabla_i F(\boldsymbol{w}_t)|^2},$$

which implies that

$$\frac{1}{T}\sum_{t=1}^{T}\mathbb{E}\left[\|\nabla F(\boldsymbol{w}_t)\|_1\right] \leq \frac{2\sqrt{3}Q}{\eta\sqrt{T}} + \sqrt{\frac{2d\delta Q}{\eta T}} + 2\sqrt{\frac{\|\boldsymbol{\sigma}\|_1 Q}{\eta}}\frac{1}{T^{1/4}}.$$

Since $Q = \mathcal{O}\left(F(\boldsymbol{w}_1) - F^* + (\eta\|\boldsymbol{\sigma}\|_1 + \eta^2\|\boldsymbol{L}\|_1)\log h(T)\right)$ and $\delta < \frac{1}{d}$, we obtain the result in Theorem 3.1.

### B.1   PROOF OF LEMMA B.2

Before we prove Lemma B.2, we first present two helper lemmas.

**Lemma B.6.** *Let $\{a_s\}_{s=1}^{\infty}$ be any sequence such that $a_s \geq 0$ for all $s$. Moreover, define $A_t = A_{t-1} + a_t$, where $A_0 = 0$. Then we have*

$$\sum_{t=1}^{T}\frac{a_t}{A_t + \delta^2} \leq \log\left(1 + \frac{A_T}{\delta^2}\right) \quad (26)$$

*Proof.* The proof is similar to (Faw et al., 2022, Lemma 15) and we repeat here for completeness. Note that for any $t \geq 1$, we have

$$\frac{a_t}{A_t + \delta^2} = 1 - \frac{A_{t-1} + \delta^2}{A_t + \delta^2} \leq \log\left(\frac{A_t + \delta^2}{A_{t-1} + \delta^2}\right).$$

The last step follows from $x \leq -\log(1 - x)$. Summing the above inequalities from $t = 1$ to $t = T$, we obtain that

$$\sum_{t=1}^{T}\frac{a_t}{A_t + \delta^2} \leq \log\left(\frac{A_T + \delta^2}{A_0 + \delta^2}\right) = \log\left(1 + \frac{A_T}{\delta^2}\right).$$

This completes the proof. $\square$

**Lemma B.7.** *Suppose that Assumption 2.4b holds and consider the update rule in AdaGrad. Then for any coordinate $i \in [d]$ and iteration $t \geq 0$, we have*

$$|\nabla_i F(\boldsymbol{w}_{t+1}) - \nabla_i F(\boldsymbol{w}_t)| \leq \eta \sqrt{L_i \|\boldsymbol{L}\|_1}. \tag{27}$$

*As a corollary, this implies that*

$$|\nabla_i F(\boldsymbol{w}_t)| \leq |\nabla_i F(\boldsymbol{w}_1)| + \eta\sqrt{L_i\|\boldsymbol{L}\|_1}t \leq \|\nabla F(\boldsymbol{w}_1)\|_\infty + \eta\sqrt{\|\boldsymbol{L}\|_\infty \|\boldsymbol{L}\|_1}t. \tag{28}$$

*Proof.* To begin with, we prove that if Assumption 2.4b holds, then for any vectors $\boldsymbol{x}, \boldsymbol{y} \in \mathbb{R}^d$,

$$\sum_{i=1}^d \frac{1}{L_i} |\nabla_i F(\boldsymbol{x}) - \nabla_i F(\boldsymbol{y})|^2 \leq \sum_{i=1}^d L_i |x_i - y_i|^2. \tag{29}$$

To see this, define the weighted Euclidean norm $\|\cdot\|_{\boldsymbol{L}}$ as $\|\boldsymbol{x}\|_{\boldsymbol{L}} := \sqrt{\sum_{i=1}^d L_i x_i^2}$ and correspondingly its dual norm is given by $\|\boldsymbol{x}\|_{\boldsymbol{L},*} := \sqrt{\sum_{i=1}^d \frac{1}{L_i} x_i^2}$. Thus, we can rewrite Assumption 2.4b as $|F(\boldsymbol{y}) - F(\boldsymbol{x}) - \langle \nabla F(\boldsymbol{x}), \boldsymbol{y} - \boldsymbol{x} \rangle| \leq \frac{1}{2} \|\boldsymbol{y} - \boldsymbol{x}\|_{\boldsymbol{L}}^2$. This is equivalent to the fact that the gradient $\nabla F(\boldsymbol{x})$ is 1-Lipschitz with respect to the norm $\|\cdot\|_{\boldsymbol{L}}$, i.e., $\|\nabla F(\boldsymbol{x}) - \nabla F(\boldsymbol{y})\|_{\boldsymbol{L},*} \leq \|\boldsymbol{x} - \boldsymbol{y}\|_{\boldsymbol{L}}$. Squaring both sides of the inequality leads to (29).

Applying (29) to the two consecutive iterates $\boldsymbol{w}_{t+1}$ and $\boldsymbol{w}_t$, we obtain that $\sum_{i=1}^d \frac{1}{L_i} |\nabla_i F(\boldsymbol{w}_{t+1}) - \nabla_i F(\boldsymbol{w}_t)|^2 \leq \sum_{i=1}^d L_i |w_{t+1,i} - w_{t,i}|^2$. Moreover, note that from the update rule of AdaGrad, it holds that

$$|w_{t+1,i} - w_{t,i}| = \eta \left| \frac{g_{t,i}}{b_{t,i} + \delta} \right| \leq \eta \left| \frac{g_{t,i}}{\sqrt{b_{t-1,i}^2 + g_{t,i}^2} + \delta} \right| \leq \eta.$$

Hence, we further have $\sum_{i=1}^d \frac{1}{L_i} |\nabla_i F(\boldsymbol{w}_{t+1}) - \nabla_i F(\boldsymbol{w}_t)|^2 \leq \eta \sum_{i=1}^d L_i = \eta \|\boldsymbol{L}\|_1$, which implies (27).

Applying the triangle inequality, we have:

$$|\nabla_i F(\boldsymbol{w}_t)| \leq |\nabla_i F(\boldsymbol{w}_1)| + \sum_{s=1}^{t-1} |\nabla_i F(\boldsymbol{w}_{s+1}) - \nabla_i F(\boldsymbol{w}_s)| \leq |\nabla_i F(\boldsymbol{w}_1)| + \eta\sqrt{L_i \|\boldsymbol{L}\|_1}t.$$

Since $|\nabla_i F(\boldsymbol{w}_1)| \leq \|\nabla F(\boldsymbol{w}_1)\|_\infty$ and $L_i \leq \|\boldsymbol{L}\|_\infty$ for any $i \in [d]$, we obtain (28). $\qquad\square$

Now we are ready to prove Lemma B.2. Recall from the definition of AdaGrad that

$$\eta_{t,i} = \frac{\eta}{\sqrt{b_{t-1,i}^2 + g_{t,i}^2} + \delta} \quad \text{and} \quad \hat{\eta}_{t,i} = \frac{\eta}{\sqrt{b_{t-1,i}^2 + \nabla_i F(\boldsymbol{w}_t)^2 + \sigma_i^2} + \delta}. \tag{30}$$

Let $a = b_{t-1,i}^2 + g_{t,i}^2$ and $b = b_{t-1,i}^2 + \nabla_i F(\boldsymbol{w}_t)^2 + \sigma_i^2$. Then

$$|\eta_{t,i} - \hat{\eta}_{t,i}| = \eta \left| \frac{1}{\sqrt{a} + \delta} - \frac{1}{\sqrt{b} + \delta} \right| = \eta \left| \frac{b - a}{(\sqrt{a} + \delta)(\sqrt{b} + \delta)(\sqrt{a} + \sqrt{b})} \right|$$

$$= \eta \left| \frac{\nabla_i F(\boldsymbol{w}_t)^2 + \sigma_i^2 - g_{t,i}^2}{(\sqrt{a} + \delta)(\sqrt{b} + \delta)(\sqrt{a} + \sqrt{b})} \right|$$

$$\leq \frac{\eta |\nabla_i F(\boldsymbol{w}_t)^2 - g_{t,i}^2| + \eta \sigma_i^2}{(\sqrt{a} + \delta)(\sqrt{b} + \delta)(\sqrt{a} + \sqrt{b})}.$$

Since $\sqrt{a} \geq |g_{t,i}|$, $\sqrt{b} \geq \max\{|\nabla_i F(\boldsymbol{w}_t)|, \sigma_i\}$, we have $|\nabla_i F(\boldsymbol{w}_t)^2 - g_{t,i}^2| \leq |\nabla_i F(\boldsymbol{w}_t) - g_{t,i}|(|\nabla_i F(\boldsymbol{w}_t)| + |g_{t,i}|) \leq |\nabla_i F(\boldsymbol{w}_t) - g_{t,i}|(\sqrt{a} + \sqrt{b})$ and $\sigma_i^2 \leq \sigma_i(\sqrt{a} + \sqrt{b})$. Therefore,

$$|\eta_{t,i} - \hat{\eta}_{t,i}| \leq \frac{\eta |\nabla_i F(\boldsymbol{w}_t) - g_{t,i}| + \eta \sigma_i}{(\sqrt{a} + \delta)(\sqrt{b} + \delta)} = \frac{1}{\eta} \left( |\nabla_i F(\boldsymbol{w}_t) - g_{t,i}| + \sigma_i \right) \eta_{t,i} \hat{\eta}_{t,i},$$

where we used $\eta_{t,i} = \frac{\eta}{\sqrt{a}+\delta}$ and $\hat{\eta}_{t,i} = \frac{\eta}{\sqrt{b}+\delta}$ in the last inequality. Hence we have,

$$|(\eta_{t,i} - \hat{\eta}_{t,i})\nabla_i F(\boldsymbol{w}_t)g_{t,i}| \leq \frac{1}{\eta}\eta_{t,i}\hat{\eta}_{t,i}(|\nabla_i F(\boldsymbol{w}_t) - g_{t,i}| + \sigma_i)|\nabla_i F(\boldsymbol{w}_t)g_{t,i}|$$

$$= \frac{\eta_{t,i}\hat{\eta}_{t,i}}{\eta}|\nabla_i F(\boldsymbol{w}_t) - g_{t,i}| \cdot |\nabla_i F(\boldsymbol{w}_t)g_{t,i}| + \frac{\sigma_i\eta_{t,i}\hat{\eta}_{t,i}}{\eta}|\nabla_i F(\boldsymbol{w}_t)g_{t,i}|.$$

Using the Cauchy-Schwartz inequality, we further have

$$\mathbb{E}\left[\eta_{t,i}\hat{\eta}_{t,i}|\nabla_i F(\boldsymbol{w}_t) - g_{t,i}| \cdot |\nabla_i F(\boldsymbol{w}_t)g_{t,i}| \mid \mathcal{F}_{t-1}\right]$$

$$\leq \hat{\eta}_{t,i}|\nabla_i F(\boldsymbol{w}_t)|\sqrt{\mathbb{E}\left[|\nabla_i F(\boldsymbol{w}_t) - g_{t,i}|^2 \mid \mathcal{F}_{t-1}\right]\mathbb{E}\left[\eta_{t,i}^2 g_{t,i}^2 \mid \mathcal{F}_{t-1}\right]}$$

$$\leq \sigma_i\hat{\eta}_{t,i}|\nabla_i F(\boldsymbol{w}_t)|\sqrt{\mathbb{E}\left[\eta_{t,i}^2 g_{t,i}^2 \mid \mathcal{F}_{t-1}\right]}$$

where the last step follows from the bounded variance in Assumption 2.3b. We proceed to bound the second term in a similar manner:

$$\mathbb{E}\left[\sigma_i\eta_{t,i}\hat{\eta}_{t,i}|\nabla_i F(\boldsymbol{w}_t)g_{t,i}| \mid \mathcal{F}_{t-1}\right] \leq \sigma_i\hat{\eta}_{t,i}|\nabla_i F(\boldsymbol{w}_t)|\sqrt{\mathbb{E}\left[\eta_{t,i}^2 g_{t,i}^2 \mid \mathcal{F}_{t-1}\right]}.$$

Combining the results, the term $\mathbb{E}\left[|(\eta_{t,i} - \hat{\eta}_{t,i})\nabla_i F(\boldsymbol{w}_t)g_{t,i}| \mid \mathcal{F}_{t-1}\right]$ is bounded as follows:

$$\mathbb{E}\left[|(\eta_{t,i} - \hat{\eta}_{t,i})\nabla_i F(\boldsymbol{w}_t)g_{t,i}| \mid \mathcal{F}_{t-1}\right] \leq \frac{2\sigma_i\hat{\eta}_{t,i}|\nabla_i F(\boldsymbol{w}_t)|}{\eta}\sqrt{\mathbb{E}\left[\eta_{t,i}^2 g_{t,i}^2 \mid \mathcal{F}_{t-1}\right]}$$

$$\leq \frac{1}{2}\hat{\eta}_{t,i}\|\nabla_i F(\boldsymbol{w}_t)\|^2 + \frac{2\hat{\eta}_{t,i}\sigma_i^2}{\eta^2}\mathbb{E}\left[\eta_{t,i}^2 g_{t,i}^2 \mid \mathcal{F}_{t-1}\right] \quad (31)$$

where we used Young's inequality in (31) in the last inequality. Finally, since $\hat{\eta}_{t,i} \leq \frac{\eta}{\sigma_i}$, we further have $\frac{\hat{\eta}_{t,i}\sigma_i^2}{\eta^2} \leq \frac{\sigma_i}{\eta}$ and this proves the inequality in (20).

Next, we prove (21) in Lemma B.2. From the definition of the step size in (30), we have:

$$\mathbb{E}\left[\sum_{t=1}^T \eta_{t,i}^2 g_{t,i}^2\right] = \eta^2\mathbb{E}\left[\sum_{t=1}^T \frac{g_{t,i}^2}{(\sqrt{b_{t-1,i}^2 + g_{t,i}^2} + \delta)^2}\right] \leq \eta^2\mathbb{E}\left[\sum_{t=1}^T \frac{g_{t,i}^2}{b_{t-1,i}^2 + g_{t,i}^2 + \delta^2}\right].$$

Using Lemma B.6, we can bound the summation with a log term as follows,

$$\eta^2\mathbb{E}\left[\sum_{t=1}^T \frac{g_{t,i}^2}{b_{t-1,i}^2 + g_{t,i}^2 + \delta^2}\right] \leq \eta^2\mathbb{E}\left[\log\left(1 + \frac{b_{T,i}^2}{\delta^2}\right)\right] \leq \eta^2\log\left(1 + \frac{\mathbb{E}\left[b_{T,i}^2\right]}{\delta^2}\right),$$

where we apply Jensen's Inequality to the concave log function in the last inequality. Moreover, since $b_{T,i}^2 = \sum_{t=1}^T g_{t,i}^2$, by using Assumptions 2.2 and 2.3b we have

$$\mathbb{E}\left[b_{T,i}^2\right] = \sum_{t=1}^T \mathbb{E}\left[g_{t,i}^2\right] \leq \sum_{t=1}^T \left(\sigma_i^2 + \mathbb{E}\left[\nabla_i F(\boldsymbol{w}_t)^2\right]\right) \leq T\|\boldsymbol{\sigma}\|_\infty^2 + \sum_{t=1}^T \mathbb{E}\left[\nabla_i F(\boldsymbol{w}_t)^2\right],$$

where we used the fact that $\sigma_i \leq \|\boldsymbol{\sigma}\|_\infty$ for any $i \in [d]$. Using the result from Lemma B.7, for any $t \in [T]$, we further have

$$\nabla_i F(\boldsymbol{w}_t)^2 \leq \left(\|\nabla F(\boldsymbol{w}_1)\|_\infty + \eta\sqrt{\|\boldsymbol{L}\|_\infty\|\boldsymbol{L}\|_1}t\right)^2 \leq \left(\|\nabla F(\boldsymbol{w}_1)\|_\infty + \eta\sqrt{\|\boldsymbol{L}\|_\infty\|\boldsymbol{L}\|_1}T\right)^2.$$

Combining all the inequalities above, we obtain that

$$\eta^2\mathbb{E}\left[\sum_{t=1}^T \frac{g_{t,i}^2}{b_{t-1,i}^2 + g_{t,i}^2 + \delta^2}\right] \leq \eta^2\log\left(1 + \frac{T\|\boldsymbol{\sigma}\|_\infty^2}{\delta^2} + \frac{T(\|\nabla F(\boldsymbol{w}_1)\|_\infty + \eta\sqrt{\|\boldsymbol{L}\|_\infty\|\boldsymbol{L}\|_1}T)^2}{\delta^2}\right)$$

Hence, we have proved the bound in (21) of Lemma B.2. This completes the proof of the results in Lemma B.2.

## C  Lower Bound Results

### C.1  Proof of Theorem 2.1

To finish the proof of Theorem 2.1, it remains to show that the function $p$ can be constructed satisfying those three conditions. This is achieved by applying the following lemma.

**Lemma C.1.** *For any given $\epsilon \in (0, \sqrt{2}]$, let $N$ be an positive integer such that $N \leq \frac{1}{\epsilon^2} + \frac{1}{2}$. Then for any $N$ points $\{x_t\}_{t=1}^N$ in $\mathbb{R}$, there exists a function $p : \mathbb{R} \to \mathbb{R}$ of one dimension such that: (i) its gradient is 1-Lipschitz; (ii) $p(x_1) - \inf p \leq 1$; (iii) $p'(x_t) = -\epsilon$ for any $t \in [N]$.*

Specifically, since $T \leq \frac{\|\boldsymbol{L}\|_\infty \Delta_f}{\epsilon^2} = \frac{1}{\tilde{\epsilon}^2}$ with $\tilde{\epsilon} = \frac{\epsilon}{\sqrt{\|\boldsymbol{L}\|_\infty \Delta_f}}$, the existence of $p$ follows from applying Lemma C.1 to the $T$ points $\{\sqrt{L_1/\Delta_f}\, x_t^{(1)}\}_{t=1}^T$.

*Proof of Lemma C.1.* We divide the proof into two cases.

**Case I:** The point $x_1$ is the largest among the $N$ points $\{x_t\}_{t=1}^N$, i.e., $x_t \leq x_1$ for any $t \in [N]$. In this case, we define the function $p : \mathbb{R} \to \mathbb{R}$ as follows;

$$p(x) = \begin{cases} -\epsilon(x - x_1), & x \in (-\infty, x_1]; \\ \frac{1}{2}(x - x_1)^2 - \epsilon(x - x_1), & x \in (x_1, +\infty). \end{cases}$$

By direct calculation, we have $p'(x) = -\epsilon$ when $x \in (-\infty, x_1]$ and $p'(x) = x - x_1 - \epsilon$ when $x \in (x_1, +\infty)$. Hence, it is straightforward to verify that $p'$ is 1-Lipschitz. Moreover, the minimum of $p$ is achieved at $x = x_1 + \epsilon$, with $\inf p = -\frac{1}{2}\epsilon^2$. Thus, we have $p(x_1) - \inf p = \frac{1}{2}\epsilon^2 \leq 1$ since $\epsilon \leq \sqrt{2}$. Finally, since $p'(x) = -\epsilon$ for all $x \leq x_1$, we conclude that $p'(x_t) = -\epsilon$ for all $t \in [N]$. Hence, the function $p$ satisfies all the three conditions in Lemma C.1.

**Case II:** There are $k$ points to the right of $x_1$ among the $N$ points $\{x_t\}_{t=1}^N$, where $1 \leq k \leq N - 1$. Since the statement in Lemma C.1 is independent of the ordering of $\{x_2, \ldots, x_N\}$, without loss of generality, we may assume that these $k$ points are $x_2, \ldots, x_{k+1}$.

We begin by defining an auxiliary function $\phi_{a,b,\epsilon}(x)$ over a given interval $[a, b]$, which is continuous, piecewise quadratic and will serve as the basic building block of our worst-case function. Specifically,

$$\phi_{a,b,\epsilon}(x) = \begin{cases} \frac{1}{2}(x - a)^2 - \epsilon(x - a), & x \in [a, \frac{a+b}{2}]; \\ -\frac{1}{2}(x - b)^2 - \epsilon(x - b) + \frac{(b-a)^2}{4} - (b-a)\epsilon, & x \in (\frac{a+b}{2}, b]. \end{cases} \tag{32}$$

Direct computation shows that $\phi_{a,b,\epsilon}'(x) = x - a - \epsilon$ for $a \leq x \leq \frac{a+b}{2}$ and $\phi_{a,b,\epsilon}'(x) = -x + b - \epsilon$ for $\frac{a+b}{2} < x \leq b$. Therefore, it is straightforward to verify that:

- $\phi_{a,b,\epsilon}(a) = 0$ and $\phi_{a,b,\epsilon}(b) = \frac{(b-a)^2}{4} - (b-a)\epsilon$;

- $\phi_{a,b,\epsilon}'$ is 1-Lipschitz and $\phi_{a,b,\epsilon}'(a) = \phi_{a,b,\epsilon}'(b) = -\epsilon$;

- $\inf_{x \in [a,b]} \phi_{a,b,\epsilon}(x) = \min\{-\frac{1}{2}\epsilon^2, \phi_{a,b,\epsilon}(b)\}$.

Having defined the function $\phi_{a,b,\epsilon}$, we now construct the function $p : \mathbb{R} \to \mathbb{R}$ as follows:

$$p(x) = \begin{cases} -\epsilon(x - x_1), & x \in (-\infty, x_1]; \\ \phi_{x_t, x_{t+1}, \epsilon}(x) + p_t, & x \in (x_t, x_{t+1}] \ (1 \leq t \leq k); \\ \frac{1}{2}(x - x_{k+1})^2 - \epsilon(x - x_{k+1}) + p_{k+1}, & x \in (x_{k+1}, +\infty). \end{cases} \tag{33}$$

Note that $p(x_t) = p_t$ and the values $\{p_t\}_{t=1}^{k+1}$ are chosen such that the function $p$ is continuous. Specifically, this requires that $\phi_{x_t, x_{t+1}, \epsilon}(x_{t+1}) + p_t = p_{t+1}$, By induction, this condition leads to

$$p_1 = 0, \ p_t = \sum_{i=1}^{t-1} \left( \frac{1}{4}(x_{i+1} - x_i)^2 - (x_{i+1} - x_i)\epsilon \right). \tag{34}$$

Now we verify that $p$ satisfies all the three conditions in Lemma C.1. First, since $p'$ is 1-Lipschitz on each interval and $p'$ is continuous, it follows that $p'$ is 1-Lipschitz over the entire real line $\mathbb{R}$. Moreover, by construction, it is straightforward to verify that $p'(x_t) = -\epsilon$ for all $t \in [k+1]$, and $p'(x) = -\epsilon$ for all $x \leq x_1$. Combining these two facts, we obtain that the third condition in Lemma C.1 is also satisfied. To verify the second condition, note that $p(x_1) = 0$. Moreover, from the definition of $p$ in (33) and the properties of $\phi_{a,b,\epsilon}$, we have

$$
p(x) \geq \begin{cases} 0, & x \in (-\infty, x_1]; \\ \min\{p_t - \frac{1}{2}\epsilon^2, p_{t+1}\}, & x \in (x_t, x_{t+1}] \ (1 \leq t \leq k); \\ p_{k+1} - \frac{1}{2}\epsilon^2, & x \in (x_{k+1}, +\infty). \end{cases}
$$

Hence, this shows that

$$
\inf p \geq \min_{t \in [k+1]} \left\{ p_t - \frac{1}{2}\epsilon^2 \right\} = \min_{t \in [k+1]} p_t - \frac{1}{2}\epsilon^2. \tag{35}
$$

Next, we provide a lower bound for $p_t$. By using Jensen's inequality, we have

$$
p_t = \sum_{i=1}^{t-1} \left( \frac{1}{4}(x_{i+1} - x_i)^2 - (x_{i+1} - x_i)\epsilon \right) = \frac{1}{4} \sum_{i=1}^{t-1} (x_{i+1} - x_i)^2 - \epsilon(x_t - x_1)
$$

$$
\geq \frac{1}{4(t-1)} \left( \sum_{i=1}^{t-1} x_{i+1} - x_i \right)^2 - \epsilon(x_t - x_1)
$$

$$
= \frac{1}{4(t-1)}(x_t - x_1)^2 - \epsilon(x_t - x_1)
$$

$$
\geq -(t-1)\epsilon^2.
$$

Since $t \leq k + 1 \leq N$, it further follows from (35) that $\inf p \geq -(N-1)\epsilon^2 - \frac{1}{2}\epsilon^2 = (-N + \frac{1}{2})\epsilon^2$. Finally, given that $N \leq \frac{1}{\epsilon^2} + \frac{1}{2}$ by assumption, we have $p(x_1) - \inf p \leq (N - \frac{1}{2})\epsilon^2 \leq 1$. Thus, we conclude that the function $p$ satisfies all the conditions in Lemma C.1. $\qquad\square$

## C.2 Proof of Theorem 3.2

We first present the following lemma, which will be used to construct the worst-case function.

**Lemma C.2.** *For any positive integer $N$, suppose that $\epsilon$ satisfies*

$$
\epsilon \leq \min \left\{ \frac{\eta \log N}{8\sqrt{N}} + \frac{1}{4\eta\sqrt{N}}, 1 \right\}. \tag{36}
$$

*Let $x_1 = 0$ and $x_t = \eta \sum_{s=1}^{t-1} \frac{1}{\sqrt{s}}$ for any $2 \leq t \leq N$. Then there exists a function $p : \mathbb{R} \to \mathbb{R}$ of one dimension such that: (i) its gradient is $1$-Lipschitz; (ii) $p(x_1) - \inf p \leq 1$; (iii) $p'(x_t) = -\epsilon$ for any $t \in [N]$.*

*Proof.* We follow a similar approach as in the proof of Lemma C.1. Specifically, we construct the function $p$ in a similar form as (33) based on the auxiliary function $\phi_{a,b,\epsilon}(x)$ defined in (32):

$$
p(x) = \begin{cases} -\epsilon(x - x_1), & x \in (-\infty, x_1]; \\ \phi_{x_t, x_{t+1}, \epsilon}(x) + p_t, & x \in (x_t, x_{t+1}] \ (1 \leq t \leq N-1); \\ \frac{1}{2}(x - x_N)^2 - \epsilon(x - x_N) + p_N, & x \in (x_N, +\infty), \end{cases}
$$

where the values $\{p_t\}_{t=1}^N$ are chosen to ensure that the function $p$ is continuous. Hence, as in (34), we have $p_1 = 0$ and

$$
p_t = \sum_{s=1}^{t-1} \left( \frac{1}{4}(x_{s+1} - x_s)^2 - (x_{s+1} - x_s)\epsilon \right) = \sum_{s=1}^{t-1} \left( \frac{\eta^2}{4s} - \frac{\eta\epsilon}{\sqrt{s}} \right), \quad \forall t \geq 2.
$$

Using the same arguments as in Lemma C.1, we can verify that $p$ has 1-Lipschitz gradient and $p'(x_t) = -\epsilon$ for all $t \in [N]$. Hence, it remains to show that $p(x_1) - \inf p \leq 1$.

To begin with, recall from (35) that $\inf p \geq \min_{t \in [N]} p_t - \frac{1}{2}\epsilon^2$, and hence our goal is to lower bound $p_t$. Moreover, note that $p_{t+1} - p_t = \frac{\eta^2}{4t} - \frac{\eta\epsilon}{\sqrt{t}}$, which implies that $p_t$ is monotonically increasing when $t \leq \frac{\eta^2}{16\epsilon^2}$ and monotonically decreasing when $t > \frac{\eta^2}{16\epsilon^2}$. It follows from this observation that $\min_{t \in [N]} p_t = \min\{p_1, p_N\}$. To lower bound $p_N$, we use the elementary inequality that $\sum_{s=1}^{N-1} \frac{1}{s} \geq \log N$ and $\sum_{s=1}^{N-1} \frac{1}{\sqrt{s}} \leq 2\sqrt{N-1} - 1 \leq 2\sqrt{N}$. This leads to

$$p_N = \frac{\eta^2}{4} \sum_{s=1}^{N-1} \frac{1}{s} - \eta\epsilon \sum_{s=1}^{N-1} \frac{1}{\sqrt{s}} \geq \frac{\eta^2}{4} \log N - 2\eta\epsilon\sqrt{N}.$$

Since $p_1 = 0$, this implies that $\inf p \geq \min\{0, \frac{\eta^2}{4} \log N - 2\eta\epsilon\sqrt{N}\} - \frac{1}{2}\epsilon^2$ and consequently

$$p(x_1) - \inf p \leq \max\left\{\frac{1}{2}\epsilon^2, 2\eta\epsilon\sqrt{N} - \frac{\eta^2}{4} \log N + \frac{1}{2}\epsilon^2\right\}.$$

Using the condition in (36), we have $\frac{1}{2}\epsilon^2 \leq \frac{1}{2} \leq 1$ and

$$2\eta\epsilon\sqrt{N} - \frac{\eta^2}{4} \log N + \frac{1}{2}\epsilon^2 \leq 2\eta\epsilon\sqrt{N} - \frac{\eta^2}{4} \log N + \frac{1}{2}$$

$$\leq 2\eta\sqrt{N}\left(\frac{\eta \log N}{8\sqrt{N}} + \frac{1}{4\eta\sqrt{N}}\right) - \frac{\eta^2}{4} \log N + \frac{1}{2} = 1.$$

Hence, we conclude that $p(x_1) - \inf p \leq 1$. $\qquad\square$

Built on Lemma C.2, we proceed to prove a complexity lower bound for AdaGrad in one dimension.

**Lemma C.3.** *Consider running AdaGrad on a one-dimensional smooth function $p$ with the scaling parameter $\eta$. For any $L > 0$ and $\Delta > 0$, there exists a function $p : \mathbb{R} \to \mathbb{R}$ and a corresponding stochastic gradient oracle such that: (i) $p$ has $L$-Lipschitz gradients and $p(x_1) - \inf p \leq \Delta$; (ii) the stochastic gradient $g_t$ is unbiased and has a bounded variance of $\sigma^2$; (iii) Given $\epsilon$ such that $\epsilon < \frac{\sqrt{L\Delta}}{16\sqrt{2}}$, if $T \leq \frac{L\Delta}{256\epsilon^2}\left(1 + \frac{\sigma^2}{4\epsilon^2}\right) \log \frac{L\Delta}{128\epsilon^2}$, then we have $\mathbb{E}\left[\min_{1 \leq t \leq T} |p'(x_t)|\right] \geq \epsilon$.*

*Proof.* We set $x_1 = 0$. To begin with, we can assume without loss of generality that $L = 1$ and $\Delta = 1$. This follows from Lemma 1 in Chewi et al. (2023), which demonstrates that if a function $p : \mathbb{R} \to \mathbb{R}$ has a 1-Lipschitz gradient and satisfies $p(0) - \inf p \leq 1$, then the rescaled function $\tilde{p}(x) = \Delta p\left(\sqrt{\frac{L}{\Delta}}x\right)$ has an $L$-Lipschitz gradient and satisfies $\tilde{p}(0) - \inf \tilde{p} \leq \Delta$. Furthermore, finding a point $\hat{x}$ such that $|\tilde{p}'(\hat{x})| \leq \epsilon$ is equivalent to finding a point $\hat{x}$ such that $|p'(\hat{x})| \leq \frac{\epsilon}{\sqrt{L\Delta}}$.

Now define $N = \frac{1}{128\epsilon^2} \log \frac{1}{128\epsilon^2}$ and we first verify that the condition in (36) is satisfied with $2\epsilon$. Specifically, we will prove that $2\epsilon \leq \sqrt{\frac{\log N}{32N}}$, which immediately implies (36) as $\frac{\eta \log N}{8\sqrt{N}} + \frac{1}{4\eta\sqrt{N}} \geq \sqrt{\frac{\log N}{32N}}$. By direct computation, we have

$$\sqrt{\frac{\log N}{32N}} = 2\epsilon\sqrt{\frac{\log N}{\log \frac{1}{128\epsilon^2}}} = 2\epsilon\sqrt{\frac{\log \frac{1}{128\epsilon^2} + \log\log \frac{1}{128\epsilon^2}}{\log \frac{1}{128\epsilon^2}}} > \epsilon,$$

where we used the fact that $\epsilon < \frac{1}{16\sqrt{2}} \Leftrightarrow \frac{1}{128\epsilon^2} > 4 \Rightarrow \log\log \frac{1}{128\epsilon^2} > 1$. Define $q_1 = 0$ and $q_t = \eta \sum_{s=1}^{t-1} \frac{1}{\sqrt{s}}$ for any $2 \leq t \leq N$. According to Lemma C.2, there exists a function $p : \mathbb{R} \to \mathbb{R}$ such that (i) its gradient is 1-Lipschitz; (ii) $p(x_1) - \inf p \leq 1$; (iii) $p'(x_t) = -2\epsilon$ for any $t \in [N]$.

Now consider running AdaGrad on the one-dimensional function $p(x)$ with the stochastic gradient oracle given by

$$\Pr(g_t = 0 \mid x_t) = \frac{\sigma^2}{\sigma^2 + 4\epsilon^2} \quad \text{and} \quad \Pr\left(g_t = \left(1 + \frac{\sigma^2}{4\epsilon^2}\right)p'(x_t) \mid x_t\right) = \frac{4\epsilon^2}{\sigma^2 + 4\epsilon^2}. \qquad (37)$$

It is straightforward to verify that $\mathbb{E}[g_t \mid x_t] = p'(x_t)$, i.e., the stochastic gradient $g_t$ is unbiased. Our goal is to show that, if $T \leq \frac{1}{256\epsilon^2}\left(1 + \frac{\sigma^2}{4\epsilon^2}\right) \log \frac{1}{128\epsilon^2} = \frac{1}{2}(1 + \frac{\sigma^2}{4\epsilon^2})N$, then we have $|p'(x_t)| = 2\epsilon$

for all $t \in [T]$ with probability at least $\frac{1}{2}$. If this is the case, we can also verify that the stochastic gradient $g_t$ has variance bounded by $\sigma^2$, and thus our construction satisfies all the required conditions.

As mentioned in the proof sketch, our key observation is the characterization of the dynamic of Ada-Grad in (10). Specifically, recall that $M_t$ denote the number of times the stochastic gradient is non-zero by time $t$ and $M_0 = 0$. By definition, we have $\mathbb{E}[M_T] = T \cdot \frac{4\epsilon^2}{\delta^2 + 4\epsilon^2}$, and thus it follows from Markov's inequality that $\Pr(M_T > 2\mathbb{E}[M_T]) \leq \frac{1}{2}$. This implies that, with probability at least $\frac{1}{2}$, we have $M_T \leq 2T \cdot \frac{4\epsilon^2}{\delta^2 + 4\epsilon^2} \leq N$. Moreover, conditioned on the event that $M_T \leq N$, we can use induction to prove that $x_t = \eta \sum_{s=1}^{M_{t-1}} \frac{1}{\sqrt{s}}$ and $p'(x_t) = -2\epsilon$ using the property of the constructed function $p$. Indeed, this holds for $t = 1$ and now suppose this holds for $t = s$. By the definition in (37), we have either $g_s = 0$ or $g_s = -2\epsilon(1 + \frac{\sigma^2}{4\epsilon^2}) = -2\epsilon - \frac{\sigma^2}{2\epsilon}$. In the former case, $M_s = M_{s-1}$ and $x_{s+1} = x_s$. In the latter case, $M_s = M_{s-1} + 1$ and $x_{s+1} = x_s + \frac{\eta}{M_s} = \sum_{j=1}^{M_{s-1}} \frac{\eta}{\sqrt{j}} + \frac{\eta}{M_s} = \sum_{j=1}^{M_s} \frac{\eta}{\sqrt{j}}$. Moreover, since $M_s \leq M_T \leq N$, we have $p'(x_{s+1}) = -2\epsilon$. Hence, in both cases, the statement holds for $t = s + 1$. Finally, using the law of total probability, we can lower bound

$$\mathbb{E}\left[\min_{1 \leq t \leq T} |p'(x_t)|\right] \geq \frac{1}{2}\mathbb{E}\left[\min_{1 \leq t \leq T} |p'(x_t)| \mid M_T \leq N\right] = \frac{1}{2} \cdot 2\epsilon.$$

This completes the proof. $\qquad\square$

Lemma C.3 states the complexity lower bound for AdaGrad for a one-dimensional function. This can be equivalently converted into a lower bound on the convergence rate, as stated in the following corollary.

**Corollary C.4.** *Consider running AdaGrad on a one-dimensional smooth function $p$ with a scaling parameter $\eta$. Then there exists a function $p_{\Delta, L, \sigma, T} : \mathbb{R} \to \mathbb{R}$ such that $p$ has $L$-Lipschitz gradient, $p(x_1) - \inf p \leq \Delta$, the stochastic gradient $g_t$ is unbiased and has a bounded variance of $\sigma^2$, and*

$$\mathbb{E}\left[\min_{1 \leq t \leq T} |p'_{\Delta, L, \sigma, T}(x_t)|\right] \geq \max\left\{\frac{1}{32}\sqrt{\frac{L\Delta \log(2T+1)}{T}}, \frac{1}{16}\left(\frac{\sigma^2 L\Delta}{T}\log\left(1 + \frac{TL\Delta}{8\sigma^2}\right)\right)^{1/4}\right\}. \tag{38}$$

*Proof.* For a given number of iterations $T$, we would like to find the largest $\epsilon$ that satisfies the condition in Lemma C.3, which serves as a valid lower bound. We will rely on the following helper lemma.

**Lemma C.5.** *Suppose $x \geq 0$. Then for $y \geq \frac{2x}{\log(x+1)}$, we have $x \leq y \log y$.*

A sufficient condition for the condition on $T$ in Lemma C.3 to satisfy is

$$2T \leq \frac{L\Delta}{128\epsilon^2}\log\frac{L\Delta}{128\epsilon^2} \quad \Leftarrow \quad \frac{L\Delta}{128\epsilon^2} \geq \frac{4T}{\log(2T+1)} \quad \Leftrightarrow \quad \epsilon \leq \sqrt{\frac{L\Delta \log(2T+1)}{512T}}.$$

Moreover, since $\sqrt{\frac{L\Delta \log(2T+1)}{1024T}} \leq \sqrt{\frac{2L\Delta T}{1024T}} \leq \sqrt{\frac{L\Delta}{512}}$, by choosing $\epsilon = \sqrt{\frac{L\Delta \log(2T+1)}{1024T}} = \frac{1}{32}\sqrt{\frac{L\Delta \log(2T+1)}{T}}$, both conditions in Lemma C.3 are satisfied. Similarly, another sufficient condition is

$$T \leq \frac{\sigma^2 L\Delta}{1024\epsilon^4}\log\frac{L\Delta}{128\epsilon^2} \quad \Leftrightarrow \quad \frac{TL\Delta}{8\sigma^2} \leq \frac{L^2\Delta^2}{2^{14}\epsilon^4}\log\frac{L^2\Delta^2}{2^{14}\epsilon^4}$$

$$\Leftarrow \quad \frac{L^2\Delta^2}{2^{14}\epsilon^4} \geq \frac{TL\Delta}{4\sigma^2}\left(\log\left(1 + \frac{TL\Delta}{8\sigma^2}\right)\right)^{-1}$$

$$\Leftrightarrow \quad \epsilon \leq \left(\frac{\sigma^2 L\Delta}{2^{14}T}\log\left(1 + \frac{TL\Delta}{8\sigma^2}\right)\right)^{1/4}.$$

Similarly, we can choose $\epsilon = \frac{1}{16}\left(\frac{\sigma^2 L\Delta}{T}\log\left(1 + \frac{TL\Delta}{8\sigma^2}\right)\right)^{1/4}$ to satisfy both conditions. Hence, we conclude that the lower bound in the corollary is satisfied. $\qquad\square$

Now we are ready to prove Theorem 3.2. As mentioned in the proof sketch, we choose the function $f : \mathbb{R}^d \to \mathbb{R}$ of the form $\sum_{i=1}^d p_{\Delta_i, L_i, \sigma_i, T}(x^{(i)})$, where $x^{(i)}$ denotes the $i$-th coordinate of $\boldsymbol{x}$ and $\Delta_i \geq 0$ with $\sum_{i=1}^d \Delta_i = \Delta_f$. By our construction, it is straightforward to verify that the function $f$ satisfies both conditions in (i) and (ii). Thus, by applying Corollary C.4 to each coordinate, we derive that

$$
\mathbb{E}\left[\min_{1\leq t\leq T}\|\nabla f(\mathbf{x}_t)\|_1\right] \geq \sum_{t=1}^T \mathbb{E}\left[\min_{1\leq t\leq T}|p'_{\Delta_i, L_i, \sigma_i, T}(x^{(i)})|\right]
$$

$$
\geq \sum_{i=1}^d C\max\left\{\sqrt{\frac{L_i\Delta_i\log T}{T}}, \left(\frac{\sigma_i^2 L_i\Delta_i}{T}\log\left(1 + \frac{TL_i\Delta_i}{\sigma_i^2}\right)\right)^{1/4}\right\},
$$

where $C$ is an absolute constant. First, consider choosing $\Delta_i = \frac{L_i\Delta_f}{\|\boldsymbol{L}\|_1}$ for all $i \in [d]$. It follows that

$$
\mathbb{E}\left[\min_{1\leq t\leq T}\|\nabla f(\mathbf{x}_t)\|_1\right] \geq \sum_{i=1}^d CL_i\sqrt{\frac{\Delta_f\log T}{\|\boldsymbol{L}\|_1 T}} = C\sqrt{\frac{\|\boldsymbol{L}\|_1\Delta_f\log T}{T}}.
$$

Second, consider choosing $\Delta_i = \frac{\sigma_i^{2/3}L_i^{1/3}}{\sum_{i=1}^d \sigma_i^{2/3}L_i^{1/3}}\Delta_f$ for $i \in [d]$. Then we have

$$
\mathbb{E}\left[\min_{1\leq t\leq T}\|\nabla f(\mathbf{x}_t)\|_1\right] \geq \sum_{i=1}^d C\left(\frac{\Delta_f\sigma_i^{8/3}L_i^{4/3}}{\sum_{i=1}^d \sigma_i^{2/3}L_i^{1/3}T}\log\left(1 + \frac{TL_i^{4/3}\Delta_f}{\sigma_i^{4/3}\sum_{i=1}^d \sigma_i^{2/3}L_i^{1/3}}\right)\right)^{1/4}
$$

$$
= C\left(\frac{(\sum_{i=1}^d \sigma_i^{2/3}L_i^{1/3})^3\Delta_f}{T}\log\left(1 + \rho T\right)\right)^{\frac{1}{4}},
$$

where $\rho = \frac{L_{\min}^{4/3}\Delta_f}{\|\boldsymbol{\sigma}\|_\infty^{4/3}\sum_{i=1}^d \sigma_i^{2/3}L_i^{1/3}}$. This completes the proof.

## C.3 Proof of Theorem 3.3

We follow a similar proof strategy as in Theorem 2.1 and use the resisting oracle argument. Consider any deterministic method $\mathcal{A}$ that has access only to a first-order oracle and let $T$ be an integer such that $T \leq \frac{\|\boldsymbol{L}\|_1\Delta_f}{\epsilon^2}$. We adversarially construct a function $f$ that satisfies the stated conditions and ensures that $\nabla f(\boldsymbol{x}_t) = \frac{1}{\|\boldsymbol{L}\|_1}[L_1\epsilon, L_2\epsilon, \ldots, L_d\epsilon] \in \mathbb{R}^d$ for any $t \in [T]$, where $\{\boldsymbol{x}_t\}_{t=1}^T$ are the queries made by $\mathcal{A}$. Note that $\|\nabla f(\boldsymbol{x}_t)\|_1 = \epsilon$ by this construction. As shown in the proof of Theorem 2.1, thanks to the deterministic nature of $\mathcal{A}$, we can simulate the algorithm using the known first-order oracle responses above and construct our function $f$ based on the queries $\{\boldsymbol{x}_t\}_{t=1}^T$.

Specifically, we construct the adversarial function $f$ of the form

$$
f(\boldsymbol{x}) = \sum_{i=1}^d \frac{L_i\Delta_f}{\|\mathbf{L}\|_1}p_i\left(\sqrt{\frac{\|\mathbf{L}\|_1}{\Delta_f}}x^{(i)}\right),
$$

where $x^{(i)}$ denotes the $i$-th coordinate of $\boldsymbol{x}$ and the one-dimensional functions $p_i : \mathbb{R} \to \mathbb{R}$ for $i \in [d]$ will be determined as follows. Fix a coordinate $i \in [d]$, let $\{x_t^{(i)}\}_{t=1}^T$ be the $i$-th coordinate of the queries $\{\boldsymbol{x}_t\}_{t=1}^T$. Since $T \leq \frac{\|\boldsymbol{L}\|_1\Delta_f}{\epsilon^2} = \frac{1}{\tilde{\epsilon}^2}$ with $\tilde{\epsilon} = \frac{\epsilon}{\sqrt{\|\boldsymbol{L}\|_1\Delta_f}}$, by invoking Lemma C.1, there exists a function $p_i$ satisfying the following conditions: (i) its gradient $p_i'$ is 1-Lipschitz; (ii) $p_i(\sqrt{\frac{\|\mathbf{L}\|_1}{\Delta_f}}x_1^{(i)}) - \inf p_i \leq 1$; (iii) $p_i'(\sqrt{\frac{\|\mathbf{L}\|_1}{\Delta_f}}x_t^{(i)}) = \tilde{\epsilon} = \frac{\epsilon}{\sqrt{\|\boldsymbol{L}\|_1\Delta_f}}$ for any $t \in [T]$. By direct computation, we can verify that $f$ satisfies Assumption 2.4b and $f(\boldsymbol{x}_1) - \inf f \leq \sum_{i=1}^d \frac{L_i\Delta_f}{\|\boldsymbol{L}\|_1} = \Delta_f$. Moreover, the $i$-th coordinate of $\nabla f(\boldsymbol{x}_t)$ is given by

$$
\frac{L_i\Delta_f}{\|\mathbf{L}\|_1}\sqrt{\frac{\|\mathbf{L}\|_1}{\Delta_f}}p_i'\left(\sqrt{\frac{\|\mathbf{L}\|_1}{\Delta_f}}x^{(i)}\right) = L_i\sqrt{\frac{\Delta_f}{\|\mathbf{L}\|_1}}\frac{\epsilon}{\sqrt{\|\boldsymbol{L}\|_1\Delta_f}} = \frac{L_i\epsilon}{\|\mathbf{L}\|_1}.
$$

Therefore, the constructed function $f$ is indeed consistent with our resisting oracle. In particular, this implies that after $\frac{\|L\|_1 \Delta_f}{\epsilon^2}$ gradient queries, Algorithm $\mathcal{A}$ fails to find a point $\hat{x}$ with $\|\nabla f(\hat{x})\|_1 < \epsilon$. This completes the proof.

### C.4 PROOF OF THEOREM 4.1

We first present a lower bound result for SGD in the one-dimensional setting. Our proof is partially inspired by (Abbaszadehpeivasti et al., 2022, Proposition 4), which studies the convergence rate of gradient descent in the noiseless setting.

**Lemma C.6.** *Consider running SGD $x_{t+1} = x_t - \eta g_t$ on a one-dimensional smooth function $p$ with a constant step size $\eta$. For any $L > 0$ and $\Delta > 0$, there exists a function $p : \mathbb{R} \to \mathbb{R}$ and a corresponding stochastic gradient oracle such that (i) $p$ has $L$-Lipschitz gradients and $p(x_1) - \inf p \leq \Delta$; (ii) the stochastic gradient $g_t$ is unbiased and has a bounded variance of $\sigma^2$; (iii) it holds that*

$$\mathbb{E}\left[\min_{1 \leq t \leq T} |p'(x_t)|\right] \geq \begin{cases} \sqrt{2L\Delta}, & \text{if } \eta \geq \frac{2}{L}; \\ \max\left\{\frac{1}{2}\sqrt{\frac{\Delta}{2\eta T + \frac{1}{2L}}}, \min\left\{\sigma\sqrt{\frac{L\eta}{2}}, \sqrt{2L\Delta}\right\}\right\}, & \text{otherwise.} \end{cases} \quad (39)$$

*Proof.* We first consider the simple case where $\eta \geq \frac{2}{L}$. Let

$$p(x) = \begin{cases} \frac{L}{2}x^2, & |x| \leq \sqrt{\frac{2\Delta}{L}}; \\ \sqrt{2L\Delta}|x| - \Delta, & |x| > \sqrt{\frac{2\Delta}{L}}, \end{cases}$$

and set the stochastic gradient oracle as the exact gradient oracle. Moreover, we initialize SGD with $x_1 = -\sqrt{\frac{2\Delta}{L}}$. It is easy to verify that both conditions (i) and (ii) are satisfied. Moreover, we can prove by induction that the iterates $x_t$ alternate between $x_1 = -\sqrt{\frac{2\Delta}{L}}$ and $x_2 = -\sqrt{\frac{2\Delta}{L}} + \eta\sqrt{2L\Delta}$. Indeed, following the update rule, we have $x_2 = x_1 - \eta p'(x_1) = -\sqrt{\frac{2\Delta}{L}} + \eta\sqrt{2L\Delta}$. Since $\eta \geq \frac{2}{L}$, it holds that $|x_2| \geq \frac{2}{L}\sqrt{2L\Delta} - \sqrt{\frac{2\Delta}{L}} = \sqrt{\frac{2\Delta}{L}}$ and hence $p'(x_2) = \sqrt{2L\Delta}$. Therefore, $x_3 = x_2 - \eta p'(x_2) = x_1$ and the repetition continues. This shows that $|p'(x_t)| = \sqrt{2L\Delta}$ for all $t \geq 1$.

For the case where $\eta < \frac{2}{L}$, we prove the lower bound by considering the following two constructions.

(i) **Construction I**: Set $\epsilon = \min\{\sigma\sqrt{\frac{L\eta}{2}}, \sqrt{2L\Delta}\}$ and without loss of generality, we initialize SGD with $x_1 = \frac{\epsilon}{L}$. Consider the function

$$p(x) = \begin{cases} \frac{L}{2}x^2, & |x| \leq \frac{\epsilon}{L}; \\ \epsilon|x| - \frac{1}{2L}\epsilon^2, & |x| > \frac{\epsilon}{L}, \end{cases} \quad (40)$$

with the stochastic gradient oracle $g(x)$ given by

$$\Pr(g(x) = 0) = \frac{\sigma^2}{\sigma^2 + \epsilon^2} \quad \text{and} \quad \Pr\left(g(x) = \left(1 + \frac{\sigma^2}{\epsilon^2}\right)p'(x)\right) = \frac{\epsilon^2}{\sigma^2 + \epsilon^2}. \quad (41)$$

It is straightforward to verify that $p(x)$ has $L$-Lipschitz gradients and $p(x_1) - \inf p \leq \frac{\epsilon^2}{2L} \leq \Delta$. Moreover, we can compute that

$$\mathbb{E}[g(x)] = \frac{\epsilon^2}{\sigma^2 + \epsilon^2}\left(1 + \frac{\sigma^2}{\epsilon^2}\right)p'(x) = p'(x),$$

$$\mathbb{E}\left[(g(x) - p'(x))^2\right] = \frac{\epsilon^2}{\sigma^2 + \epsilon^2}\left(1 + \frac{\sigma^2}{\epsilon^2}\right)^2 p'(x)^2 - p'(x)^2 = \frac{\sigma^2}{\epsilon^2}p'(x)^2.$$

Since $|p'(x)| \leq \epsilon$ for any $x \in \mathbb{R}$, this further implies that $\mathbb{E}\left[(g(x) - p'(x))^2\right] \leq \sigma^2$. Thus, the first two conditions in Lemma C.6 are satisfied. Finally, we will prove by induction

that the iterates $\{x_t\}_{t=1}^T$ alternate between the two points $\frac{\epsilon}{L}$ and $\frac{\epsilon}{L} - \eta\left(\epsilon + \frac{\sigma^2}{\epsilon}\right)$ and the gradient norm at both points is $\epsilon$. This is clearly true for $t = 1$. Now suppose this holds for $t = s$. We consider the following scenarios:

- Assume that $x_s = \frac{\epsilon}{L}$, then $p'(x_s) = \epsilon$ and by the construction in (41) we have either $g_s = 0$ or $g_s = (1 + \frac{\sigma^2}{\epsilon^2})\epsilon = \epsilon + \frac{\sigma^2}{\epsilon}$. In the former case, we have $x_{s+1} = x_s = \frac{\epsilon}{L}$, while in the latter case we have $x_{s+1} = x_s - \eta\left(\epsilon + \frac{\sigma^2}{\epsilon}\right) = \frac{\epsilon}{L} - \eta\left(\epsilon + \frac{\sigma^2}{\epsilon}\right)$. Hence, the statement holds for $t = s + 1$.

- Otherwise, assume that $x_s = \frac{\epsilon}{L} - \eta\left(\epsilon + \frac{\sigma^2}{\epsilon}\right)$. Since $\epsilon \leq \sigma\sqrt{\frac{L\eta}{2}}$, this implies that $\sigma^2 \geq \frac{2\epsilon^2}{L\eta}$ and thus $\frac{\epsilon}{L} - \eta\left(\epsilon + \frac{\sigma^2}{\epsilon}\right) \leq \frac{\epsilon}{L} - \frac{\eta\sigma^2}{\epsilon} \leq -\frac{\epsilon}{L}$. According to (40), we have $p'(x_s) = -\epsilon$ and thus $g_s = 0$ or $g_s = -\epsilon - \frac{\sigma^2}{\epsilon}$. Similarly, we can show that the statement continues to hold in both cases.

(ii) **Construction II:** Set $\epsilon = \frac{1}{2}\sqrt{\frac{\Delta}{2\eta T + \frac{1}{2L}}}$ and we initialize SGD with $x_1 = 0$. Similar to the proof of Theorem 2.1, we will construct our function based on $\phi_{a,b,\epsilon}(x)$ defined in (32). Specifically, let $N = 2T \cdot \frac{4\epsilon^2}{\sigma^2 + 4\epsilon^2} = \frac{\Delta - 2\epsilon^2/L}{\eta(4\epsilon^2 + \sigma^2)}$ and define the $N$ points as and $q_t = (t-1)\eta\left(2\epsilon + \frac{\sigma^2}{2\epsilon}\right)$ for $t \in [N]$. Then consider the function

$$p(x) = \begin{cases} -2\epsilon x, & x \in (-\infty, 0]; \\ L\phi_{q_t, q_{t+1}, 2\epsilon/L}(x) + p_t, & x \in (q_t, q_{t+1}] \ (1 \leq t \leq N-1); \\ \frac{L}{2}(x - q_N)^2 - 2\epsilon(x - q_N) + p_N, & x \in (q_N, +\infty), \end{cases}$$

where the values $\{p_t\}_{t=1}^N$ are determined to ensure that the function $p$ is continuous. Specifically, this requires $p_1 = 0$ and $p_{t+1} = p_t + L\phi_{q_t, q_{t+1}, 2\epsilon/L}(q_{t+1}) = p_t + \frac{L}{4}(q_{t+1} - q_t)^2 - 2\epsilon(q_{t+1} - q_t)$, which leads to

$$p_{t+1} = t\left(\frac{L\eta^2}{4}\left(2\epsilon + \frac{\sigma^2}{2\epsilon}\right)^2 - \eta(4\epsilon^2 + \sigma^2)\right) \geq -\eta t(4\epsilon^2 + \sigma^2).$$

Moreover, we set the stochastic gradient oracle as

$$\Pr(g(x) = 0) = \frac{\sigma^2}{\sigma^2 + 4\epsilon^2} \quad \text{and} \quad \Pr\left(g(x) = \left(1 + \frac{\sigma^2}{4\epsilon^2}\right)p'(x)\right) = \frac{4\epsilon^2}{\sigma^2 + 4\epsilon^2}. \quad (42)$$

Again, it is straightforward to verify that $p'$ is $L$-Lipschitz, and due to the definition of $\phi$ in (32), it holds that $p'(q_t) = -2\epsilon$ for all $t \in [N]$. Now we will show that $p(x_1) - \inf p \leq \Delta$. To see this, note that similar to the arguments in Lemma C.1, one can show that

$$\inf p = \min_{t \in [N]} p_t - \frac{2}{L}\epsilon^2 \geq -\eta(N-1)(4\epsilon^2 + \sigma^2) - \frac{2}{L}\epsilon^2 \geq -\Delta.$$

As a result, we obtain $p(x_1) - \inf p \leq \Delta$.

Finally, we will show that $\mathbb{E}\left[\min_{1 \leq t \leq T+1} |p'(x_t)|\right] \geq \epsilon$. Our strategy is similar to the proof of Lemma C.3. Let $M_t$ denote the number of times the stochastic gradient is non-zero by time $t$ and set $M_0 = 0$. Then from the definition of the stochastic gradient oracle in (41), we have $\mathbb{E}[M_T] = \frac{4\epsilon^2}{\sigma^2 + 4\epsilon^2}T$. By Markov's inequality, we have $\Pr(M_T > 2\mathbb{E}[M_T]) \leq \frac{1}{2}$. This implies that, with probability at least $\frac{1}{2}$, we have $M_T \leq 2T\frac{4\epsilon^2}{\sigma^2 + 4\epsilon^2} = N$. Conditioned on the event that $M_T \leq N$, we can use induction to prove that $x_t = M_{t-1}\eta\left(2\epsilon + \frac{\sigma^2}{2\epsilon}\right)$ and $p'(x_t) = -2\epsilon$ for all $t \in [T]$. This is true for $t = 1$ and suppose that this holds for $t = s$. By the definition in (42), we have either $g_s = 0$ or $g_s = -2\epsilon - \frac{\sigma^2}{2\epsilon}$. In the former case, $M_s = M_{s-1}$ and $x_{s+1} = x_s = M_s\eta\left(2\epsilon + \frac{\sigma^2}{2\epsilon}\right)$. In the latter case, $M_s = M_{s-1} + 1$ and $x_{s+1} = x_s - \eta g_s = (M_{s-1} + 1)\eta\left(2\epsilon + \frac{\sigma^2}{2\epsilon}\right) = M_s\eta\left(2\epsilon + \frac{\sigma^2}{2\epsilon}\right)$. Moreover, Since

$M_s \le N$, we also have $p'(x_{s+1}) = -2\epsilon$. Hence, in both cases, the statement continues to hold for $t = s + 1$. Using the law of total probability, we can lower bound

$$\mathbb{E}\left[\min_{1 \le t \le T} |p'(x_t)|\right] \ge \frac{1}{2}\mathbb{E}\left[\min_{1 \le t \le T} |p'(x_t)| \mid M_T \le N\right] = \frac{1}{2} \cdot 2\epsilon = \epsilon.$$

This completes the proof.

Since both constructions provide a valid lower bound, we can take the maximum of the two as the final lower bound. This leads to Lemma C.6. $\square$

Now we are ready to prove Theorem 4.1. Denote by $p_{\Delta,L,\sigma,\eta,T}(\cdot)$ the function in Lemma C.6 that achieves the lower bound. Consider the function

$$f(\boldsymbol{x}) = \sum_{i=1}^{d} p_{\Delta/d, L_i, \sigma_i, \eta, T}(x^{(i)}),$$

where $x^{(i)}$ denotes the $i$-th coordinate of the vector $\boldsymbol{x}$. If $\eta \ge \frac{2}{\|\boldsymbol{L}\|_\infty}$, then it follows from the first lower bound in Lemma C.6 that

$$\mathbb{E}\left[\min_{1 \le t \le T} \|\nabla f(\boldsymbol{x}_t)\|_1\right] \ge \sqrt{\frac{2\|\boldsymbol{L}\|_\infty \Delta}{d}}.$$

If $\eta < \frac{2}{\|\boldsymbol{L}\|_\infty} \le \frac{1}{L_i}$ for all $i \in [d]$, it follows from the second lower bound in Lemma C.6 that :

$$\mathbb{E}\left[\min_{1 \le t \le T} \|\nabla f(\boldsymbol{x}_t)\|_1\right] \ge \sum_{i=1}^{d} \mathbb{E}\left[\min_{1 \le t \le T} |p'_{\Delta/d, L_i, \sigma_i, \eta, T}(x_t^{(i)})|\right]$$

$$\ge \sum_{i=1}^{d} \max\left\{\frac{1}{2}\sqrt{\frac{\Delta/d}{2\eta T + \frac{1}{2L_i}}}, \min\left\{\sigma_i\sqrt{\frac{L_i\eta}{2}}, \sqrt{2L_i\frac{\Delta}{d}}\right\}\right\}$$

$$\ge \sum_{i=1}^{d} \frac{1}{4}\sqrt{\frac{\Delta/d}{2\eta T + \frac{1}{2L_i}}} + \sum_{i=1}^{d} \frac{1}{2}\min\left\{\sigma_i\sqrt{\frac{L_i\eta}{2}}, \sqrt{2L_i\frac{\Delta}{d}}\right\} \quad (43)$$

$$\ge \frac{1}{4}\sqrt{\frac{d\Delta}{2\eta T + \frac{1}{2L_{\min}}}} + \sum_{i=1}^{d} \frac{1}{2}\min\left\{\sigma_i\sqrt{\frac{L_i\eta}{2}}, \sqrt{2L_i\frac{\Delta}{d}}\right\}. \quad (44)$$

Now we would like to establish a lower bound that is independent of the step size $\eta$. Let $L_{\min} = \min_{i \in [d]} L_i$. We consider the following cases.

(i) If $2\eta T \le \frac{1}{2L_{\min}}$, then the lower bound in (44) is at least $\frac{1}{4}\sqrt{\frac{d\Delta}{2\eta T + \frac{1}{2L_{\min}}}} \ge \frac{1}{4}\sqrt{L_{\min}d\Delta}$.

(ii) If $2\eta T \ge \frac{1}{2L_{\min}}$ but $\sigma_i\sqrt{\frac{L_i\eta}{2}} \ge \sqrt{2L_i\frac{\Delta}{d}}$ for some $i \in [d]$, then the lower bound in (44) is at least $\frac{1}{2}\sqrt{\frac{2L_i\Delta}{d}} \ge \frac{1}{2}\sqrt{\frac{2L_{\min}\Delta}{d}}$.

(iii) Finally, If $2\eta T \ge \frac{1}{2L_{\min}}$ and $\sigma_i\sqrt{\frac{L_i\eta}{2}} < \sqrt{2L_i\frac{\Delta}{d}}$ for all $i \in [d]$, then the lower bound in (44) becomes

$$\frac{1}{4}\sqrt{\frac{d\Delta}{2\eta T + \frac{1}{2L_{\min}}}} + \sum_{i=1}^{d} \frac{1}{2}\sigma_i\sqrt{\frac{L_i\eta}{2}} \ge \frac{1}{8}\sqrt{\frac{d\Delta}{\eta T}} + \frac{1}{2\sqrt{2}}\sum_{i=1}^{d}\sigma_i\sqrt{L_i}\sqrt{\eta}.$$

Since $\eta < \frac{2}{\|\boldsymbol{L}\|_\infty}$, we can further lower bound the above inequality by $\frac{1}{8}\sqrt{\frac{d\Delta}{\eta T}} \ge \frac{1}{8}\sqrt{\frac{d\|\boldsymbol{L}\|_\infty\Delta}{2T}}$. Moreover, by using the elementary inequality $a + b \ge 2\sqrt{ab}$ for any $a, b \ge 0$, we also obtain that

$$\frac{1}{8}\sqrt{\frac{d\Delta}{\eta T}} + \frac{1}{2\sqrt{2}}\sum_{i=1}^{d}\sigma_i\sqrt{L_i}\sqrt{\eta} \ge \frac{d^{1/4}\Delta_f^{1/4}(\sum_{i=1}^{d}\sigma_i\sqrt{L_i})^{1/2}}{4 \cdot 2^{1/4}T^{1/4}}.$$

Hence, in this case we have

$$\mathbb{E}\left[\min_{1\le t\le T}\|\nabla f(\boldsymbol{x}_t)\|_1\right] \ge \max\left\{\frac{1}{8}\sqrt{\frac{d\|\boldsymbol{L}\|_\infty\Delta}{2T}}, \frac{d^{1/4}\Delta_f^{1/4}(\sum_{i=1}^d \sigma_i\sqrt{L_i})^{1/2}}{4\cdot 2^{1/4}T^{1/4}}\right\}$$

$$\ge \frac{1}{16}\sqrt{\frac{d\|\boldsymbol{L}\|_\infty\Delta}{2T}} + \frac{d^{1/4}\Delta_f^{1/4}(\sum_{i=1}^d \sigma_i\sqrt{L_i})^{1/2}}{8\cdot 2^{1/4}T^{1/4}}$$

By taking the minimum of all three cases, we conclude that

$$\mathbb{E}\left[\min_{1\le t\le T}\|\nabla f(\boldsymbol{x}_t)\|_1\right] \ge \min\left\{\frac{1}{16}\sqrt{\frac{d\|\boldsymbol{L}\|_\infty\Delta}{2T}} + \frac{d^{1/4}\Delta_f^{1/4}(\sum_{i=1}^d \sigma_i\sqrt{L_i})^{1/2}}{8\cdot 2^{1/4}T^{1/4}}, \frac{1}{4}\sqrt{\frac{L_{\min}\Delta}{d}}\right\}.$$

Note that the second term in our lower bound is a constant independent of $T$. Thus, when $T$ is sufficiently large, we obtain the result in Theorem 4.1.

