# OpenReview forum: "Convergence Analysis of Adaptive Gradient Methods under Refined Smoothness and Noise Assumptions"
_ICLR.cc/2025/Conference — Submitted to ICLR 2025_

### Official Review · Reviewer_HgUb · 2024-10-26

**Soundness:** 3
**Presentation:** 3
**Contribution:** 3
**Rating:** 5
**Confidence:** 3

**Summary:**

This paper explores whether the iteration complexity of AdaGrad surpasses that of SGD in stochastic non-convex optimization. By introducing refined assumptions that align with the coordinate-wise nature of adaptive gradients, the authors use the L1-norm of the gradient as a stationarity measure and demonstrate that AdaGrad's iteration complexity outperforms SGD by a factor of d.

**Strengths:**

1. The paper successfully bridges a gap in the theoretical understanding of AdaGrad compared to SGD in non-convex settings.
2. The comparison between the upper bound for AdaGrad and the lower bound for SGD is well-executed, clearly highlighting the advantage of AdaGrad.

**Weaknesses:**

1. The refined assumptions used are not particularly novel; similar approaches have been explored in previous works. Additionally, the use of the L1-norm for the theoretical analysis has been discussed in recent literature, limiting the paper's overall novelty.
2. The authors have already highlighted the main differences between this paper and [1], noting that [1] examines the upper bound for SGD, while this paper focuses on the lower bound. However, despite this distinction, the two papers remain quite similar. A more in-depth discussion of the unique techniques or specific analytical steps that differentiate this work would enhance the clarity and contribution of the paper.
3. The lack of experimental validation is a significant shortcoming. While the paper emphasizes theoretical analysis, some empirical results would be beneficial. Demonstrating that AdaGrad reduces the bound of SGD by a factor of d through experiments—especially by varying the value of d—would greatly support the theoretical claims.

[1] Yuxing Liu, Rui Pan, and Tong Zhang. Large batch analysis for adagrad under anisotropic smoothness. arXiv preprint arXiv:2406.15244, 2024.

**Questions:**

See the Weaknesses above.

---

> ### Author Response · Authors · 2024-11-19
>
> We thank the reviewer for their constructive feedback.
>
> **Q1 The refined assumptions used are not particularly novel; similar approaches have been explored in previous works. Additionally, the use of the L1-norm for the theoretical analysis has been discussed in recent literature, limiting the paper's overall novelty.**
>
> **A1**
> Thank you for your feedback. We acknowledge that the refined assumptions and the use of the $\ell_1$-norm have been explored in prior works.
> However, the key novelty of our paper lies in our complexity lower bound results. To the best of our knowledge, this is the first work to establish lower bounds in terms of the $\ell_1$-norm while explicitly highlighting the dimensional dependence in the convergence rate. Moreover, our approach is distinct from standard techniques for deriving lower bounds in optimization, such as those based on the zero-chain constructions [Carmon et al., 2020; Arjevani et al., 2023], which focus on dimension-free lower bounds.
> By directly comparing the lower bound for SGD with the upper bound for AdaGrad, we are also able to establish a provable advantage of AdaGrad over SGD in the non-convex setting, which, to our knowledge, is the first in the literature.
>
> Yair Carmon, John C Duchi, Oliver Hinder, and Aaron Sidford. Lower bounds for finding stationary
> points I. Mathematical Programming, 2020.
>
> Yossi Arjevani, Yair Carmon, John C Duchi, Dylan J Foster, Nathan Srebro, and Blake Woodworth.
> Lower bounds for non-convex stochastic optimization. Mathematical Programming, 2023.
>
> -----
>
> **Q2 The authors have already highlighted the main differences between this paper and [1], noting that [1] examines the upper bound for SGD, while this paper focuses on the lower bound. However, despite this distinction, the two papers remain quite similar. A more in-depth discussion of the unique techniques or specific analytical steps that differentiate this work would enhance the clarity and contribution of the paper.**
>
> **A2**
> Thank you for your question.
> First, we would like to note that paper [1] appeared on arXiv *two weeks after* the initial version of our paper was online. While there are some overlapping contributions, we consider [1] to be concurrent and independent work. Therefore, it should not be viewed as a weakness of our paper.
>
> Specifically, as we discussed in our introduction, the authors of [1] considered the convergence of AdaGrad under anisotropic smoothness and noise assumptions, similar to our refined assumptions 2.3b and 2.4b. They also established an upper bound on AdaGrad’s convergence in terms of the gradient’s $\ell_1$-norm, comparable to our Theorem 3.1.
>
> A key distinction, however, lies in how the two works compare AdaGrad and SGD.
> To demonstrate AdaGrad's advantages over SGD, the authors in [1] rely on the classical upper bound for SGD using $\ell_2$-norm under standard smoothness and noise assumptions. This may not be the most fair comparison,
> as a direct analysis of SGD in terms of $\ell_1$-norm with refined assumptions could potentially yield a stronger bound. Essentially, they are comparing an __upper bound__ for AdaGrad against an __upper bound__ for SGD, which could be loose. In contrast,  our work provides a __lower bound__ for SGD, allowing us to directly compare AdaGrad’s upper bound with SGD’s lower bound to demonstrate a clear advantage for AdaGrad. Moreover, we validate the tightness of our AdaGrad upper bound through two lower specific bounds: one tailored to AdaGrad and another applicable to deterministic first-order methods.
>
> ----
>
> **Q3 The lack of experimental validation is a significant shortcoming. While the paper emphasizes theoretical analysis, some empirical results would be beneficial. Demonstrating that AdaGrad reduces the bound of SGD by a factor of d through experiments—especially by varying the value of d—would greatly support the theoretical claims.**
>
> **A3**
> Thank you for your suggestion.
> While additional numerical experiments would provide valuable insights, our focus in this paper is on establishing theoretical foundations for the advantage of coordinate-wise adaptive methods over SGD. Our results provide provable guarantees for when and why these methods outperform SGD and AdaGrad-norm under specific conditions, which aligns with existing empirical observations in the literature. We believe this theoretical framework sufficiently supports our claims, and we encourage future empirical work to explore these conditions in greater detail.

---

> > ### Comment · Reviewer_HgUb · 2024-11-27
> >
> > Thank you for addressing my concerns. I acknowledge that this paper presents novel theoretical analyses and results. The only minor weakness is that aligning the experiments more closely with the theoretical findings would strengthen the paper. However, considering the overall contributions, I am inclined to raise my score. Best.

---

### Official Review · Reviewer_RTtE · 2024-10-26

**Soundness:** 3
**Presentation:** 3
**Contribution:** 3
**Rating:** 5
**Confidence:** 2

**Summary:**

This paper provided a convergence analysis of the adaptive gradient (AdaGrad) in the nonconvex setting. Relaxing standard assumptions of Lipschitz gradients and bounded noise variance in the $\ell_2$ metric, where the SGD is worst-case optimal in terms of finding a near-stationary point, the paper proposed a coordinate-wise analog conditions under which the AdaGrad can improve the dimension $d$ dependence in the iteration complexity over the SGD under the $\ell_1$ norm. In particular, the paper proved: (i) an upper bound of AdaGrad; (ii) a lower bound of SGD; and (iii) a lower bound of AdaGrad (matching AdaGrad upper bound for the noiseless term). The paper showed that for certain problem configurations, the gap between the AdaGrad upper bound and the lower bound of SGD is non-trivial. More precisely, the iteration complexity of AdaGrad outperforms SGD by a factor of $d$, thus arguing the improved theoretical guarantee of AdaGrad over SGD in certain nonconvex settings.

**Strengths:**

The results of this paper are mainly theoretical. This paper compares the convergence rate of AdaGrad with SGD under coordinate-wise
assumptions (smoothness and noise) and $\ell_1$-norm stationarity measure. The paper demonstrates a provable advantage of AdaGrad over SGD in certain nonconvex settings.

The paper is well-written and the intuition for introducing the $\ell_1$-norm of the gradient as the stationarity measure is well-explained.

The paper mainly compared the upper bound for AdaGrad with the lower bound of SGD, while the derivation of the latter result is missing in the concurrent work [1], as stated by the authors.

[1] Yuxing Liu, Ru Pan, and Tong Zhang. Large batch analysis for Adagrad under anisotropic smoothness. arXiv preprint arXiv:2406.15244,2024.

**Weaknesses:**

It would be better if the comparison between SGD and AdaGrad in convex settings is provided, for both $\ell_2$-norm measure (I guess it belongs to the existing literature) and $\ell_1$-norm stationary measure. Moreover, even though the paper is theoretical, it would still be beneficial to demonstrate the gain of dimension dependence using some numeric experiments.

**Questions:**

The improvement of AdaGrad over SGD is demonstrated with abstract assumptions on smoothness and noise. Can the paper give a concrete example or more primitive assumptions (such as in the Abstract motivation of training neural networks) when those proposed assumptions are satisfied?

In the non-convex setting, this paper utilizes the $\ell_1$-norm of the gradient as the stationarity measure and compares the convergence rate. When comparing different optimization algorithms, they might converge to different stationary points (such as due to initialization and implicit bias). How does the initialization impact the AdaGrad dynamics? Is there any discussion on the works for the implicit bias of SGD and AdaGrad in non-convex settings?

As raised in the Weakness, it would be beneficial to demonstrate the gain of dimension dependence using some numeric experiments, i.e., the gap between the AdaGrad upper bound and SGD lower bound, and their difference in iteration complexity when changing dimensions.

---

> ### Author Response · Authors · 2024-11-19
>
> We thank the reviewer for their constructive feedback.
>
> **Q1 It would be better if the comparison between SGD and AdaGrad in convex settings is provided, for both $\ell_2$-norm measure (I guess it belongs to the existing literature) and $\ell_1$-norm stationary measure.**
>
> **A1** Thank you for the suggestion. In the convex setting, a common performance metric is the function value gap, $F(\mathbf{w}) - F^*$. It has been shown that depending on the geometry of the feasible set and the sparsity of the gradients, AdaGrad's convergence bound can be either better or worse than that of SGD by a factor of $\sqrt{d}$, where $d$ is the problem's dimension. For more detailed discussions, we refer the reviewer to (Hazan, 2016; Orabona 2019).  As for comparisons based on the gradient $\ell_2$-norm and $\ell_1$-norm in the convex setting, to our knowledge, this has not been explored in the literature. While our convergence results would also apply in this setting, they may not yield the tightest bounds.
>
> ----
>
> **Q2 The improvement of AdaGrad over SGD is demonstrated with abstract assumptions on smoothness and noise. Can the paper give a concrete example or more primitive assumptions (such as in the Abstract motivation of training neural networks) when those proposed assumptions are satisfied?**
>
> **A2** Thank you for your suggestion. We note that our assumptions are relatively standard and have been adopted in prior works such as (Bernstein et al., 2018), which also provided empirical validation for Assumption 2.3b. However, as acknowledged in their paper, verifying Assumption 2.4b in practical neural network training problems can be challenging.
>
> On the other hand, it is not hard to construct simple examples that satisfy the refined assumptions. For instance, consider the quadratic function $F(\mathbf{w}) = \sum_{i=1}^d (\frac{1}{2}L_i (w^{(i)})^2 + b_i w^{(i)})$, where $L_i>0$ and $b_i$ are scalars, and $w^{(i)}$ denotes the $i$-th coordinate of $\mathbf{w} \in \mathbb{R}^d$. We can verify that this function satisfies Assumption 2.4b. Moreover, suppose the stochastic gradient $\mathbf{g}_t$ is given by $\mathbf{g}_t = \nabla F(\mathbf{w}_t) + \mathbf{n}_t$, where $\mathbf{n}_t$ is the stochastic noise satisfying $\mathbb{E} [({n}_t^{(i)})^2] \leq \sigma_i^2$. Under these conditions, Assumption 2.3b is also satisfied.
>
> -------
>
> **Q3 In the non-convex setting, this paper utilizes the $\ell_1$-norm of the gradient as the stationarity measure and compares the convergence rate. When comparing different optimization algorithms, they might converge to different stationary points (such as due to initialization and implicit bias). How does the initialization impact the AdaGrad dynamics? Is there any discussion on the works for the implicit bias of SGD and AdaGrad in non-convex settings?**
>
> **A3** Thank you for your suggestion. In non-convex optimization, it is indeed possible for different optimization algorithms to converge to different stationary points, leading to potentially different generalization properties.
> There are several recent works on the implicit bias of AdaGrad and SGD, including (Qian and Qian, 2019; Wang et al., 2021; Lyu and Li, 2020). However, our focus in this paper is on the optimization perspective, where we consider all stationary points equivalently, and exploring implicit bias lies beyond the scope of our current work.
>
> Qian, Qian, and Xiaoyuan Qian. "The implicit bias of AdaGrad on separable data." NeurIPS 2019.
>
> Wang, B., Meng, Q., Chen, W., and Liu, T. Y. "The implicit bias for adaptive optimization algorithms on homogeneous neural networks." ICML 2021.
>
> Lyu, Kaifeng, and Jian Li. "Gradient Descent Maximizes the Margin of Homogeneous Neural Networks." ICLR 2020.
>
> ------
>
> **Q4 Even though the paper is theoretical, it would be beneficial to demonstrate the gain of dimension dependence using some numeric experiments, i.e., the gap between the AdaGrad upper bound and SGD lower bound, and their difference in iteration complexity when changing dimensions.**
>
> **A4** Thank you for your suggestion.
> While additional numerical experiments would provide valuable insights, our focus in this paper is on establishing theoretical foundations for the advantage of coordinate-wise adaptive methods over SGD. Our assumptions are also relatively standard and have been adopted in prior work, such as (Bernstein et al., 2018).  Our results provide provable guarantees for when and why these methods outperform scalar SGD and AdaGrad-norm under specific conditions, which aligns with existing empirical observations in the literature. We believe this theoretical framework sufficiently supports our claims, and we encourage future empirical work to explore these conditions in greater detail.

---

> > ### Author Response · Authors · 2024-11-28
> >
> > Thank you again for taking the time to review our work! We wanted to check if our responses have addressed your concerns or if you have any additional feedback or suggestions for us. Your input is invaluable, and we greatly appreciate your effort and thoughtful review.

---

### Official Review · Reviewer_STji · 2024-10-28

**Soundness:** 3
**Presentation:** 3
**Contribution:** 3
**Rating:** 6
**Confidence:** 4

**Summary:**

The paper provides expected average $\ell_{1}$-norm convergence upperbound and lowerbound for AdaGrad under coordinate-wise bounded variance gradient noise and coordinate-wise smoothness assumptions for non-convex objectives. Comparisons between coordinate-wise AdaGrad and SGD are performed to demonstrate dimensional advantage of coordinate-wise AdaGrad over scalar SGD under favorable conditions.

**Strengths:**

• Interesting fine-grained analysis that shows advantage of coordinate-wise methods over scalar versions via better dimensional dependency.

• The lower bound for AdaGrad is interesting where the implication is that the log(T) factor in AdaGrad is shown to be not improve-able.

**Weaknesses:**

- Comparisons are not sound.
    - When comparing with SGD, especially when the lower bound utilizes new coordinate-wise smoothness, one should also consider a coordinate wise version of SGD as a baseline i.e. SGD with stepsize $\eta_{i}=\Theta(1/L_{i})$ for each $i\in[d]$. In that case, both algorithms are allowed $O(d)$ memory (rather than vanilla SGD only uses O(1) memory). There, I think the benefit of adaptive methods will be more of automatically adapting to these coordinate wise smoothness rather than any dimensional improvement in the convergence rate.
    - The worst-case comparison on lines 358-362 is not clear: For example, if $\sigma_{i}=1$ for all $i$, then $\|\|\sigma\|\|_2=\sqrt{d}$ and $\|\|\sigma\|\|_1=d$. There, the dimensional dependency is still $d$ and not $\sqrt{d}$. What am I missing?
    - Previous work [Liu et al 2023] provides $\ell_{1}$-norm convergence in high probability for sub-Gaussian coordinate-wise noise for AdaGrad. There, I don't think it's fair to compare results in high-probability with results converge in expectation (in the related works section and Appendix A).
    - The comparison between AdaGrad and SGD on lines 512-516 is not too convincing. The authors should also comment on the setting when AdaGrad is worse than SGD and discuss the reason behind that.

- Upperbound analysis for AdaGrad is not novel and upperbound results seem incremental.
    - The analysis seems quite similar to Ward et al 2020 with the generalization to coordinate-wise version of the variance and smoothness. Furthermore, $\ell_{1}$-norm convergence for non-convex objective under coordinate-wise noise is not new (e.g. see Liu et al. 2023), and the extension of Liu et al. 2023's proof to coordinate-wise smoothness seems not too difficult.
    - Could the authors comment on the main difficulties needed to generalize existing in expectation results to use the new assumptions?

- Need to verify new assumptions in practical settings:
    - The authors should show that these assumptions are observed in real world tasks (e.g. training transformers, CNNs, etc.) where adaptive methods converge faster than SGD, Adagrad-norm.
    - The authors should perform controlled synthetic experiments to verify that coordinate-wise adaptive methods outperform scalar SGD and AdaGrad-norm under the conditions posed in the comparison.

**Questions:**

The concerns from weakness section should be addressed. I am willing to raise my score.

To make the paper's contribution more significant, I think the authors should consider performing some empirical validation of the new assumptions (coordinate noise, sparsity rate, coordinate smoothness, etc.), especially in modern DNNs settings (e.g. LLMs) where dimension is large and perform similar analysis on a wider range of algorithms like Adam or RMSProp.

---

> ### Author Response · Authors · 2024-11-19
>
> We thank the reviewer for their constructive feedback.
>
> **Q1 When comparing with SGD, especially when the lower bound utilizes new coordinate-wise smoothness, one should also consider a coordinate wise version of SGD as a baseline i.e. SGD with stepsize $\eta_i = \Theta(1/L_i)$ for each $i \in [d]$.**
>
> **A1** Thank you for your suggestion. We agree that the coordinate-wise version of SGD could serve as an idealized baseline for AdaGrad. However, this algorithm is impractical because the coordinate-wise Lipschitz constants $L_i$ are typically unknown. If we treat these coordinate-wise step sizes as hyperparameters and rely on manual tuning, this process becomes intractable as the number of the hyperparameters scales with the problem's dimension. Hence, to our knowledge, unlike the vanilla SGD, such coordinate-wise SGD has not been successfully used for practical problems. We also wish to emphasize that our focus is to demonstrate the advantage of AdaGrad over SGD in the non-convex setting, which remains a challenging open problem [Chen & Hazan, 2024]. While we agree that the ability to automatically adapt to Lipschitz constants is a key advantage of AdaGrad, its coordinate-wise nature is also essential to understanding its gains over SGD.
>
> ----
>
> **Q2 The worst-case comparison on lines 358-362 is not clear: For example, if $\sigma_i = 1$ for all $i$, then $\\|\sigma\\|_2 = \sqrt{d}$ and $\\|\sigma\\|_1 =d$. There, the dimensional dependency is still $d$ and not $\sqrt{d}$. What am I missing?**
>
> **A2** We apologize for the confusion. In this discussion, our aim is to present our convergence bounds under the standard assumptions (i.e., Assumption 2.3a and Assumption 2.4a), to facilitate the comparison with existing results. In the worst case, this yields the bound $\\tilde{\\mathcal{O}}\\Bigl(\\sqrt{\\frac{d\\|\mathbf{L}\\|\_{\\infty}\\Delta\_F}{T}} + \sqrt{d}\left({\frac{{\\|\boldsymbol{\sigma}\\|^{2}\_{2} {\\|\mathbf{L}\\|\_{\infty}\Delta_F}}}{{T}}} \right)^{\frac{1}{4}} + \frac{{\sqrt{d}\\|\boldsymbol{\sigma}\\|\_2}}{T^{\frac{1}{4}}}\Bigr)$. Therefore, if we focus on the explicit dependence on the dimension, ignoring $\\|\mathbf{L}\\|_{\infty}$,  $\\|\boldsymbol{\sigma}\\|_2$ and $\Delta_F$ as is common in prior work, this bound simplifies to $\tilde{\mathcal{O}}(\frac{\sqrt{d}}{\sqrt{T}} + \frac{\sqrt{d}}{T^{1/4}})$. We will clarify this point further in the revision.
>
> ----
>
> **Q3 Previous work [Liu et al. 2023] provides $\ell_1$-norm convergence in high probability for sub-Gaussian coordinate-wise noise for AdaGrad. There, I don't think it's fair to compare results in high probability with results converge in expectation (in the related works section and Appendix A).**
>
> **A3** Thank you for pointing this out. We will revise our paper to clarify that the results in [Liu et al., 2023] are high-probability convergence results and thus not directly comparable to the in-expectation results presented in our work.
>
> -----
>
> **Q4 The comparison between AdaGrad and SGD on lines 512-516 is not too convincing. The authors should also comment on the setting when AdaGrad is worse than SGD and discuss the reason behind that.**
>
> **A4**
> We would like to clarify that our comparison on lines 512-516 focuses on the **upper complexity bound** of AdaGrad versus the **lower complexity bound** of SGD. Specifically, we showed that when the vector $\boldsymbol{\sigma}$ is sparse ($\phi(\boldsymbol{\sigma}) \approx 1/d$) and the noise variance vector is aligned with the smoothness vector ($R \approx 1$), the upper bound of AdaGrad is below the lower bound of SGD, establishing a provable advantage for AdaGrad. Conversely, when $\phi(\boldsymbol{\sigma}) \approx 1$ and $R \approx 0$, the upper bound of AdaGrad is worse than the lower bound of SGD; however, this does not necessarily mean that AdaGrad performs worse than SGD, as neither bound may be exactly tight.

---

> > ### Author Response · Authors · 2024-11-19
> >
> > **Q5 The analysis seems quite similar to Ward et al 2020 with the generalization to coordinate-wise version of the variance and smoothness. Furthermore, $\ell_1$-norm convergence for non-convex objective under coordinate-wise noise is not new (e.g. see Liu et al. 2023), and the extension of Liu et al. 2023's proof to coordinate-wise smoothness seems not too difficult.**
> >
> > **A5**
> > Thank you for your question. Our convergence analysis indeed draws inspiration from the "decorrelated step size" technique introduced by Ward et al. (2020). However, Ward et al. (2020) rely on the additional assumption that the gradient norm, $\\|\nabla F(\mathbf{w})\\|$, is uniformly bounded to establish lower bounds on decorrelated step sizes (see Section 3.2.1 of their paper). In contrast, our analysis takes a different approach to remove this restrictive assumption, as detailed in Steps 2 and 3 of the proof sketch.
> >
> > Compared with (Liu et al. 2023), our convergence analysis introduces several technical differences, particularly in Step 2, where we establish lower bounds on the auxiliary step sizes $\tilde{\eta}_{T,i}$. We would be happy to elaborate further if the reviewer finds it necessary. Also, as discussed in Appendix A, Liu et al. (2023) provide high-probability convergence results that incur additional dependence on the dimension, which may make it difficult to establish a clear advantage for AdaGrad over SGD.
> >
> > We would also like to emphasize that our key novelty lies in providing accompanying lower bound results, including one specific to SGD, one specific to AdaGrad, and one applicable to general deterministic first-order methods. By comparing the upper bound of AdaGrad with the lower bound of SGD, we establish the first provable advantage of AdaGrad in the non-convex setting.
> >
> > -------
> >
> > **Q6 Could the authors comment on the main difficulties needed to generalize existing in expectation results to use the new assumptions?**
> >
> > **A6** Thank you for your question. We note that existing in-expectation results primarily focus either on AdaGrad-Norm or on analyzing AdaGrad using the gradient $\ell_2$-norm as the stationarity measure.
> >
> > Compared to those works on AdaGrad-Norm, a key challenge in our analysis is combining the per-coordinate bounds to derive a tight upper bound on an appropriate gradient norm. This is particularly evident in Eq. (3). For AdaGrad-Norm, since the same step size is used across all coordinates, we have $\hat{\eta}\_{t,i} = \hat{\eta}\_t$ for all $i \in [d]$. In this case, the left-hand side of (3) simplifies to $\sum_{t=1}^T \hat{\eta}_t \Vert \nabla F(\mathbf{w}_t) \Vert_2^2$, leading to an upper bound on the $\ell_2$-norm of the gradient. However, in AdaGrad, the step sizes for each coordinate are updated independently, making it unclear how the quantity on the left-hand side of (3) can be related to the norm of the gradient which we seek to upper bound.
> >
> > Regarding prior works on AdaGrad, they often exhibit explicit dependence on the problem’s dimension,  partly due to their choice of the $\ell_2$-norm. One of our key challenges is to eliminate this explicit dimensional dependence and instead express the convergence bound in terms of the vectors $\mathbf{L}$ and $\boldsymbol{\sigma}$, which facilitate the comparison with SGD.
> >
> > ------
> >
> > **Q7 The authors should show that these assumptions are observed in real world tasks (e.g. training transformers, CNNs, etc.) where adaptive methods converge faster than SGD, Adagrad-norm. The authors should perform controlled synthetic experiments to verify that coordinate-wise adaptive methods outperform scalar SGD and AdaGrad-norm under the conditions posed in the comparison.**
> >
> > **A7** Thank you for your suggestion.
> > While additional numerical experiments would provide valuable insights, our focus in this paper is on establishing theoretical foundations for the advantage of coordinate-wise adaptive methods over SGD. Our assumptions are also relatively standard and have been adopted in prior work, such as (Bernstein et al., 2018).  Our results provide provable guarantees for when and why these methods outperform scalar SGD and AdaGrad-norm under specific conditions, which aligns with existing empirical observations in the literature. We believe this theoretical framework sufficiently supports our claims, and we encourage future empirical work to explore these conditions in greater detail.
> >
> > ------
> >
> > **Q8 The authors should consider performing similar analysis on a wider range of algorithms like Adam or RMSProp.**
> >
> > **A8** Thank you for your suggestion. Given the close relationship between Adam, RMSProp, and AdaGrad, we anticipate that our analysis could be extended to encompass these methods. However, such an extension lies beyond the scope of this paper, and we leave it as a direction for future work.

---

> > ### Comment · Reviewer_STji · 2024-11-24
> >
> > Thank you for addressing my concerns. Some confusions still remain and I would appreciate if the authors could help me further clarify.
> >
> > > Hence, to our knowledge, unlike the vanilla SGD, such coordinate-wise SGD has not been successfully used for practical problems.
> >
> > This work is primarily theoretical and I think the coordinate-wise SGD step size is of theoretical interest i.e. to find out the true benefits of adaptive methods. I think the author should still perform the comparison due to theoretical relevance.
> >
> > >  In the worst case, this yields the bound $\tilde{\mathcal{O}}\Bigl(\sqrt{\frac{d\|\mathbf{L}\|_{\infty}\Delta_F}{T}} + \sqrt{d}\left({\frac{{\|\boldsymbol{\sigma}\|^{2}_{2} {\|\mathbf{L}\|_{\infty}\Delta_F}}}{{T}}} \right)^{\frac{1}{4}} + \frac{{\sqrt{d}\|\boldsymbol{\sigma}\|_2}}{T^{\frac{1}{4}}}\Bigr)$. Therefore, if we focus on the explicit dependence on the dimension, ignoring $\|\mathbf{L}\|_{\infty}$, $\|\boldsymbol{\sigma}\|_2$ and $\Delta_F$ as is common in prior work, this bound simplifies to $\tilde{\mathcal{O}}(\frac{\sqrt{d}}{\sqrt{T}} + \frac{\sqrt{d}}{T^{1/4}})$.
> >
> > I don't understand nor agree why we could ignore $\|\mathbf{L}\|_{\infty}$, $\|\boldsymbol{\sigma}\|_2$ and $\Delta_F$ since each of them, especially the noise term $\sigma$ could definitely depend on the dimension e.g. see my example.
> >
> > > We would like to clarify that our comparison on lines 512-516 focuses on the upper complexity bound of AdaGrad versus the lower complexity bound of SGD. Specifically, we showed that when the vector $\boldsymbol{\sigma}$ is sparse ($\phi(\boldsymbol{\sigma}) \approx 1/d$) and the noise variance vector is aligned with the smoothness vector ($R \approx 1$), the upper bound of AdaGrad is below the lower bound of SGD, establishing a provable advantage for AdaGrad.
> >
> > Similarly to the previous concern, there's the third term for the upperbound for AdaGrad that is not considered in that paragraph and I'm not sure how the comparison is fair without considering all the terms that could affect the final dimensional dependency. It would be great if the authors could make the comparison more explicit.

---

> > > ### Author Response · Authors · 2024-11-26
> > >
> > > We thank the reviewer for the insightful questions. Below we address your additional concerns.
> > >
> > > -----------
> > >
> > > **Q1 This work is primarily theoretical and I think the coordinate-wise SGD step size is of theoretical interest i.e. to find out the true benefits of adaptive methods. I think the author should still perform the comparison due to theoretical relevance.**
> > >
> > > **A1** Thank you for your insightful comment. We will include coordinate-wise SGD as an idealized baseline for theoretical comparison in the revised version of the paper. Moreover, we would like to clarify that our primary goal is not to design the fastest optimization algorithms under the refined smoothness and noise assumptions. Instead, our focus is on establishing the theoretical advantages of adaptive methods over SGD—a phenomenon that is widely observed in practice but lacks theoretical understanding in the non-convex setting.
> > >
> > > ------------
> > >
> > >
> > > **Q2 I don't understand nor agree why we could ignore $\\|\mathbf{L}\\|_{\infty}$, $\\|\boldsymbol{\sigma}\\|_2$ and $\Delta_F$ since each of them, especially the noise term could definitely depend on the dimension e.g. see my example.**
> > >
> > > **A2** Thank you for pointing this out. We agree that these problem parameters may implicitly depend on the problem's dimension. However, it is conventional in the optimization literature to treat them as constants when simplifying bounds. For instance, under the standard smoothness and noise assumptions, vanilla SGD achieves a convergence rate of $O(\sqrt{\frac{\\|\mathbf{L}\\|\_{\infty} \Delta_F}{T}} + \left(\frac{\\|\mathbf{L}\\|\_{\infty}\Delta_F \\||\boldsymbol{\sigma}\\|\_2^2}{T}\right)^{1/4})$ in terms of the expected gradient $\ell_2$-norm. This is often referred to as a "dimensional-independent" rate, despite the potential implicit dependence of the parameters on the dimension.
> > >
> > > That said, we understand your concern, and to avoid potential confusion, we will revise the paragraph in question and retain the explicit dependence on $\\|\mathbf{L}\\|_{\infty}$, $\\|\boldsymbol{\sigma}\\|_2$ and $\Delta_F$ in our bounds.
> > >
> > > -----------------------
> > >
> > > **Q3 Similarly to the previous concern, there's the third term for the upper bound for AdaGrad that is not considered in that paragraph and I'm not sure how the comparison is fair without considering all the terms that could affect the final dimensional dependency. It would be great if the authors could make the comparison more explicit.**
> > >
> > > **A3** Thank you for raising this point. As mentioned in Line 506-507, when the noise is relatively small, i.e., $\\||\boldsymbol{\sigma}\\|_1 \ll \sqrt{\\|\mathbf{L}_1\\| \Delta_F}$, the third term in AdaGrad's upper bound will be dominated by the first two terms. For this reason, we omitted the third term when comparing the upper bound of AdaGrad with the lower bound of SGD. We will state this assumption more explicitly in the revised paper.

---

> > > > ### Comment · Reviewer_STji · 2024-11-27
> > > >
> > > > Dear authors,
> > > > Thank you for addressing my concerns. I have raised my score and hope the authors could incorporate these clarifications in the revision of the paper.

---

### Official Review · Reviewer_aRGb · 2024-11-04

**Soundness:** 4
**Presentation:** 4
**Contribution:** 3
**Rating:** 6
**Confidence:** 3

**Summary:**

The paper studies the convergence of AdaGrad under: 1) coordinate-wise smoothness and noise variance, and 2) convergence of the $L_1$ norm. In the noiseless setting, AdaGrad is proven to be superior (not inferior) to SGD in terms of constant dependence. Moreover, in the stochastic setting where the noise is sparse and aligns with the smoothness parameter, AdaGrad also exhibits better constant dependence. This is done by also establishing a lower bound for SGD when using the $L_1$ norm. Furthermore, the paper presents lower bounds for AdaGrad and all deterministic first-order algorithms.

**Strengths:**

- The paper is well-written and easy to follow, with a well-organized language.
- The technical contribution is solid, presenting a novel perspective on showing the provable advantage of AdaGrad under coordinate-wise smoothness and $L_1$-norm convergence.
- Additionally, the lower bounds provided are novel in the context of $L_1$-norm convergence.

**Weaknesses:**

- The provable advantage is clear in the deterministic setting, while for the stochastic setting, the benefit depends, it could be even worse. In real-world applications, it is unknown whether the noise is sparse and aligns with the smoothness parameters.

**Questions:**

- It’s unclear why $L_1$-norm convergence is considered a better measure, if not to artificially distinguish between SGD and AdaGrad. Since adaptive methods generally exhibit faster convergence in practice, is there a rationale behind why the $L_1$-norm result aligns more closely with the observations?
- Can we achieve similar advantages if SGD employs a coordinate-wise stepsize, such as $1/(L_i \sqrt{T})$ for the $i$-th coordinate?

---

> ### Author Response · Authors · 2024-11-19
>
> We thank the reviewers for their insightful feedback.
>
> **Q1 The provable advantage is clear in the deterministic setting, while for the stochastic setting, the benefit depends, it could be even worse. In real-world applications, it is unknown whether the noise is sparse and aligns with the smoothness parameters.**
>
> **A1** In the stochastic setting, the comparison between the upper bound of AdaGrad and the lower bound of SGD depends on two key quantities: $\phi(\boldsymbol{\sigma}) = \frac{\\|\boldsymbol{\sigma}\\|^2_1}{d \\|\boldsymbol{\sigma}\\|^2_2}$, which measures the density of the vector $\boldsymbol{\sigma}$,  and  $R = \frac{\sum_{i=1}^d \sigma_i \sqrt{L_i}}{\|\boldsymbol{\sigma}\|_2 \sqrt{\\|\mathbf{L}\\|_1}}$, which represents the cosine similarity between the vectors $[\sigma_1, \dots, \sigma_d] \in \mathbb{R}^d$ and $[\sqrt{L_1}, \dots, \sqrt{L_d}] \in \mathbb{R}^d$. When the vector $\boldsymbol{\sigma}$ is sparse ($\phi(\mathbf{\boldsymbol{\sigma}}) \approx 1/d$) and the two vectors $[\sigma_1, \dots, \sigma_d]$ and $[\sqrt{L_1}, \dots, \sqrt{L_d}]$ are aligned ($R \approx 1$), AdaGrad reduces the bound of SGD by a factor of $d$. Conversely, when $\phi(\mathbf{\boldsymbol{\sigma}}) \approx 1$ and $R \approx 0$, our upper bound for AdaGrad can indeed be worse than the lower bound of SGD, but this does not imply AdaGrad performs worse than SGD.
>
> In addition, we note that AdaGrad is not expected to outperform SGD in all scenarios. For instance, in the standard online convex optimization setting, it is known that under certain geometry of the feasible set and gradient density conditions, the rate of AdaGrad can be worse than SGD by a factor of $\sqrt{d}$. To the best of our knowledge, our work is the first to identify a specific scenario in which AdaGrad demonstrates a provable advantage over SGD in a non-convex setting.
>
> ---
>
> **Q2 It’s unclear why $L_1$-norm convergence is considered a better measure, if not to artificially distinguish between SGD and AdaGrad. Since adaptive methods generally exhibit faster convergence in practice, is there a rationale behind why the $L_1$-norm result aligns more closely with the observations?**
>
> **A2** This is a good question. We acknowledge that the choice of the $\ell_1$-norm is primarily motivated by theoretical considerations, and in practice, one may still observe a performance gain of AdaGrad over SGD when the gradient $\ell_2$-norm is used. However, formalizing this improvement using $\ell_2$-norm theoretically is challenging, as we explain below.
>
> This difficulty stems from the use of a **worst-case complexity analysis** (which is standard in optimization literature), and the selection of a stationarity metric affects the class of "worst-case functions" considered. As shown in Theorem 2.1, when the gradient $\ell_2$-norm is chosen as the stationarity metric, the worst-case function for any deterministic first-order method is effectively one-dimensional. Intuitively, SGD and AdaGrad achieve a similar performance on such functions, and thus the advantage of AdaGrad will not be reflected in the wrost-case complexity bound. Conversely, when the gradient $\ell_1$-norm is used as the stationarity metric, the worst-case function is a $d$-dimensional function with an imbalance across the coordinates (see Theorem 4.1), where it is necessary to make progress in all coordinates to achieve low loss. In this setting, the coordinate-wise nature of AdaGrad allows it to outperform SGD as established in our paper.
>
> One might intuitively argue that the worst-case function in the $\ell_1$-norm setup aligns more closely with practical loss functions, as achieving a low loss requires the algorithm to make progress across all coordinates. However, formally verifying this alignment remains challenging, and we leave it as a direction for future work.
>
> ---
>
> **Q3 Can we achieve similar advantages if SGD employs a coordinate-wise stepsize, such as $1/(L_i \sqrt{T})$ for the $i$-th coordinate?**
>
> **A3** Good question. In theory, using coordainte-wise step sizes of $\eta_i = 1/(L_i \sqrt{T})$ in SGD would lead to a convergence rate comparable to AdaGrad. However, this algorithm is impractical because the coordinate-wise Lipschitz constants $L_i$ are typically unknown. If we treat these coordinate-wise step sizes as hyperparameters and rely on manual tuning, this process becomes intractable as the number of the hyperparameters scales with the problem's dimension. In contrast, AdaGrad automatically adjusts the step size for each coordinate based on past observed stochastic gradients, requiring minimal tuning effort.

---

> > ### Author Response · Authors · 2024-11-28
> >
> > Thank you again for taking the time to review our work and for your positive comments! We just wanted to check if our responses have addressed your concerns or if you have any additional feedback or suggestions for us. Your input is invaluable, and we greatly appreciate your effort and thoughtful review.

---

### Public Comment · ~Kyunghun_Nam1 · 2024-12-06
**Some question in your proof process**

As someone working on a similar topic in mathematical optimization, I found your paper very interesting and insightful.
However, I encountered a part in the middle that I did not fully understand, and I have a question.

On page 6, equation $(5)$, you state that you used the following inequality, but I believe this statement is mathematically incorrect in its current form.

For example, consider the random variables $(X, Y)$:
$(X, Y) = (1, 1)$, $(X, Y) = (2, 2)$ with probability $1/2$.

In this case, $\mathbb{E}[X^2/Y] = 1.5$ while $\mathbb{E}[X^2] / \mathbb{E}[Y] = 1.6667$.
This serves as a counterexample.

That statement should be modified to hold under the condition that $X$ and $Y$ are \textbf{independent} positive random variables. Could you clarify this point?

---

### Meta-Review · Area_Chair_A5w9 · 2024-12-22

**Metareview:**

This paper considers adaptive gradient methods (AdaGrad) under coordinate-wise smoothness and bounded noise variance assumptions, in non-convex settings. Authors provide convergence guarantees and comparisons between AdaGrad and SGD and demonstrate advantage of coordinate-wise AdaGrad over SGD.


This paper was reviewed by four reviewers and received the following Scores/Confidence: 5/2, 6/4, 6/3, 5/3. I think paper is studying an interesting topic but authors are not able to convince the reviewers sufficiently well about the separation between the algorithms they study. The following concerns were brought up by the reviewers:

- Unclear comparisons: Authors provide various comparisons throughout the paper that it is difficult to justify their fairness. The comparisons could be under different scenarios each of which may influence the derived conclusion. This part needs clarification.

- Novelty of the analysis was brought up by reviewers, to be limited. In particular, the upper bounds derived in this paper are not too hard to derive from existing work.

- Reviewers also found the conditions rather opaque and hard to verify in practical settings.


No reviewers championed the paper and they are not particularly excited about the paper.
I think majority of the concerns can be addressed but that would require significant revision and another set of reviews. As such, based on the reviewers' suggestion, as well as my own assessment of the paper, I recommend not including this paper to the ICLR 2025 program.

**Additional Comments On Reviewer Discussion:**

Authors's response addressed some of the concerns brought up by the reviewers. However, reviewers were still not convinced strongly in favor of this work.

---

### Decision · Program_Chairs · 2025-01-22

Reject